# Estradiol elicits distinct firing patterns in arcuate nucleus kisspeptin neurons of females through altering ion channel conductances

Jian Qiu[1]*, Margaritis Voliotis[2,3], Martha A Bosch[1], Xiao Feng Li[4], Larry S Zweifel[5,6], Krasimira Tsaneva-Atanasova[2,3], Kevin T O'Byrne[4], Oline K Rønnekleiv[1,7], Martin J Kelly[1,7]*

[1]Department of Chemical Physiology and Biochemistry, Oregon Health & Science University, Portland, United States; [2]Department of Mathematics and Statistics, University of Exeter, Exeter, United Kingdom; [3]Living Systems Institute, University of Exeter, Exeter, United Kingdom; [4]Department of Women and Children's Health, School of Life Course and Population Sciences, King's College London, London, United Kingdom; [5]Department of Psychiatry and Behavioral Sciences, University of Washington, Seattle, United States; [6]Depatment of Pharmacology, University of Washington, Seattle, United States; [7]Division of Neuroscience, Oregon National Primate Research Center, Beaverton, United States

*For correspondence:
qiuj@ohsu.edu (JQ);
kellym@ohsu.edu (MJK)

Competing interest: The authors declare that no competing interests exist.

## eLife Assessment

This **valuable** study combined multiple approaches to gain insight into why rising estradiol levels, by influencing hypothalamic neurons, ultimately lead to ovulation. The experimental data were **solid**, but evidence for the conclusion that the findings explain how estradiol acts in the intact female were incomplete because they lacked experimental conditions that better approximate physiological conditions. Nevertheless, the work will be of interest to reproductive biologists working on ovarian biology and female fertility.

**Abstract** Hypothalamic kisspeptin (Kiss1) neurons are vital for pubertal development and reproduction. Arcuate nucleus Kiss1 (Kiss1[ARH]) neurons are responsible for the pulsatile release of gonadotropin-releasing hormone (GnRH). In females, the behavior of Kiss1[ARH] neurons, expressing Kiss1, neurokinin B (NKB), and dynorphin (Dyn), varies throughout the ovarian cycle. Studies indicate that 17β-estradiol (E2) reduces peptide expression but increases *Slc17a6* (*Vglut2*) mRNA and glutamate neurotransmission in these neurons, suggesting a shift from peptidergic to glutamatergic signaling. To investigate this shift, we combined transcriptomics, electrophysiology, and mathematical modeling. Our results demonstrate that E2 treatment upregulates the mRNA expression of voltage-activated calcium channels, elevating the whole-cell calcium current that contributes to high-frequency burst firing. Additionally, E2 treatment decreased the mRNA levels of canonical transient receptor potential (TPRC) 5 and G protein-coupled K⁺ (GIRK) channels. When *Trpc5* channels in Kiss1[ARH] neurons were deleted using CRISPR/SaCas9, the slow excitatory postsynaptic potential was eliminated. Our data enabled us to formulate a biophysically realistic mathematical model of Kiss1[ARH] neurons, suggesting that E2 modifies ionic conductances in these neurons, enabling the transition from high-frequency synchronous firing through NKB-driven activation of TRPC5 channels to a short bursting mode facilitating glutamate release. In a low E2 milieu, synchronous firing of Kiss1[ARH]

neurons drives pulsatile release of GnRH, while the transition to burst firing with high, preovulatory levels of E2 would facilitate the GnRH surge through its glutamatergic synaptic connection to preoptic Kiss1 neurons.

## Introduction

Hypothalamic kisspeptin (Kiss1) neurons and its cognate receptor (GPR 54 or Kiss1 R) are essential for pubertal development and reproduction, and may also be involved in the control of energy homeostasis (*Kotani et al., 2001*; *de Roux et al., 2003*; *Seminara et al., 2003*; *Messager et al., 2005*; *Shahab et al., 2005*; *d'Anglemont de Tassigny et al., 2008*; *Qiu et al., 2018*; *Rønnekleiv et al., 2022*). Kisspeptin neurons within the arcuate nucleus of the hypothalamus (Kiss1$^{ARH}$) co-express Kiss1, neurokinin B (NKB), and dynorphin (Dyn), which are all downregulated by 17β-estradiol (E2) (*Goodman et al., 2007*; *Navarro et al., 2009*). However, Kiss1$^{ARH}$ neurons also express vesicular glutamate transporter 2 (*Vglut2*) and release glutamate, and both *Slc17a6* expression and glutamate release are upregulated by E2 in females (*Qiu et al., 2018*). This indicates that peptides and glutamate in Kiss1$^{ARH}$ neurons are differently modulated by E2 and suggests that there is a complex E2 regulation in these neurons such that they transition from predominantly peptidergic to glutamatergic neurotransmission and hence from a 'pulsatile' to 'surge' mode. It has been known for decades that neurons located within the arcuate nucleus are responsible for the pulsatile release of GnRH and subsequently pulsatile release of LH from the pituitary gland (*O'Byrne et al., 1991*; *Moenter et al., 1993*). In this respect, it is now generally accepted that Kiss1$^{ARH}$ neurons are the main neurons responsible for the generation of pulsatile LH release, but the underlying cellular conductances generating this activity have not been elucidated.

Morphological studies have provided evidence that Kiss1$^{ARH}$ neurons can communicate directly with each other (*Lehman et al., 2010*; *Navarro et al., 2009*; *Navarro et al., 2011*). Furthermore, Kiss1$^{ARH}$ neurons express the NKB receptor, TacR3, as well as the kappa (κ) opioid receptor (KOR), whereas GPR54 is not expressed in Kiss1$^{ARH}$ neurons, rendering them unresponsive to kisspeptin (*d'Anglemont de Tassigny et al., 2008*; *Navarro et al., 2009*; *Wakabayashi et al., 2010*). Using optogenetics and whole-cell recordings, we demonstrated that high-frequency photoactivation of Kiss1$^{ARH}$ neurons induces a NKB-mediated slow excitatory postsynaptic potential (EPSP), which is mediated by the recruitment of canonical transient receptor potential (TRCP5) channels. The release of NKB is limited by co-released dynorphin, which acts presynaptically to inhibit further release. Together, the two peptides cause synchronized firing of Kiss1$^{ARH}$ neurons (*Qiu et al., 2016*; *Kelly et al., 2018*; *Qiu et al., 2021*), whereas kisspeptin and glutamate appear to be the main output signals from Kiss1$^{ARH}$ neurons (*Qiu et al., 2016*; *Qiu et al., 2018*; *Voliotis et al., 2021*; *Liu et al., 2021*).

Although single-action potential-generated calcium influx is sufficient to trigger the release of classical neurotransmitters such as glutamate, high-frequency firing (10–20 Hz) is required for the release of neuropeptides such as kisspeptin, NKB, and dynorphin (*Qiu et al., 2016*). Indeed, the slow EPSP, which underlies the synchronization, is similar to the 'plateau potential' that has been described in hippocampal and cortical neurons (*Zhang et al., 2011*; *Arboit et al., 2020*). Many neurons, including Kiss1$^{ARH}$ neurons, express the biophysical properties that allow them to continue to persistently fire even after a triggering synaptic event has subsided (*Zylberberg and Strowbridge, 2017*; *Qiu et al., 2016*). Moreover, the intrinsic bi-stability of neurons that generates persistent firing activity has been linked to a calcium-activated, non-selective cation current ($I_{CAN}$) (*Zylberberg and Strowbridge, 2017*), and TRPC channels, specifically TRPC5 channels, are thought to be responsible for the $I_{CAN}$ in cortical neurons (*Zhang et al., 2011*). Therefore, we postulated that TacR3 activation via NKB drives influx of $Ca^{+2}$ through TRPC5 channels, leading to greater build-up of $[Ca^{2+}]_i$ that facilitates the opening of more TRPC5 channels in a self-sustaining manner. Indeed, using the fast intracellular calcium chelator BAPTA, which has been shown to robustly inhibit TRPC5 channel activation in heterologous cells (*Blair et al., 2009*), we have been able to abolish the slow EPSP and persistent firing in Kiss1$^{ARH}$ neurons following optogenetic stimulation in female mice (*Qiu et al., 2021*).

Although the expression of peptides in Kiss1$^{ARH}$ neurons is downregulated by high circulating levels (late follicular levels) of E2, the intrinsic excitability of Kiss1$^{ARH}$ neurons and the glutamate release by Kiss1$^{ARH}$ neurons are increased by *Cacna1g* (Cav3.1, T-type calcium channel), *Hcn1* and *Hcn2* (hyperpolarization-activated, cyclic nucleotide-gated channels) mRNA expression and *Slc17a6* mRNA

expression, respectively (*Qiu et al., 2018*). Burst firing in CNS neurons, which efficiently releases fast amino acid transmitters like glutamate, is generated primarily by the T-type calcium channel current ($I_T$) (e.g., in thalamic relay neurons), and the rhythmicity of this burst firing is dependent on the h current ($I_h$) (for review, see *Zagotta and Siegelbaum, 1996*; *Lüthi and McCormick, 1998*). $I_h$ is mediated by the HCN channel family, which includes channel subtypes 1–4, of which *Hcn1* and *Hcn2* are the main channels in Kiss1$^{ARH}$ neurons. $I_h$ depolarizes neurons from hyperpolarized states, raising the membrane potential into the range of $I_T$ activation (*Erickson et al., 1993a*; *Erickson et al., 1993b*; *Kelly and Rønnekleiv, 1994*; *Lüthi and McCormick, 1998*; *Zhang et al., 2009*). $I_T$ is mediated by the low voltage-activated (LVA) calcium channels, $Ca_v3.1$–3.3 (for review, see *Perez-Reyes, 2003*). $I_T$ initiates a transient $Ca^{2+}$-driven depolarization above the threshold for action potential initiation (i.e., a low threshold spike) (*Tsien et al., 1987*; *Llinás, 1988*). This depolarization then drives neurons to fire an ensemble (burst) of $Na^+$-driven action potentials.

Based on the above compelling evidence, we postulated that Kiss1$^{ARH}$ neurons transition from peptidergic neurotransmission, driving the pulsatile release of GnRH via kisspeptin release into the median eminence, to glutamatergic transmission that facilitates in the preovulatory surge of GnRH (*Lin et al., 2021*; *Shen et al., 2022*) through their projection to the anteroventral periventricular/ periventricular nucleus Kiss1 (Kiss1$^{AVPV/PeN}$) neurons (*Qiu et al., 2016*). Therefore, we initiated studies to thoroughly characterize the effects of high circulating (late follicular) levels of E2 on the expression of the full complement of voltage-activated calcium channels (and currents) and the opposing $K^+$ channels, involved in the repolarization, on the excitability of Kiss1$^{ARH}$ neurons. We performed whole-cell recordings and single-cell RT-PCR analysis of Kiss1$^{ARH}$ neurons to determine which channels are involved in the physiological transition from peptidergic to glutamatergic neurotransmission. Our physiological findings were incorporated into a mathematical model that accounts for the E2 effects on the firing activity of Kiss1$^{ARH}$ neurons and validates our hypothesis that high levels of E2 facilitate the transition from peptidergic to glutamatergic neurotransmission.

## Results
### Voltage-activated Ca$^{2+}$ channels contribute to burst firing of Kiss1$^{ARH}$ neurons

Our whole-cell current-clamp recordings of Kiss1$^{ARH}$ neurons revealed that there is a significant transition of these Kiss1 neurons from a silent/tonic firing mode in the ovariectomized state to bursting/ irregular firing with E2 treatment (*Figure 1*, *Figure 1—source data 1*). Therefore, we sought to elucidate the cationic channels contributing to this physiological transition. Previously, we showed that an increase in the intracellular calcium concentrations can potentiate TRPC5 channel current in POMC neurons (*Qiu et al., 2010*). Additionally, chelating intracellular calcium with BAPTA abolishes the slow EPSP and persistent firing in Kiss1$^{ARH}$ neurons (*Qiu et al., 2021*). Hence, to investigate the contributions of voltage-activated Ca$^{2+}$ ahannels to the increase in intracellular calcium, we measured the peak calcium current contributed by both the low and high voltage-activated calcium channels. The inward currents evoked by the voltage pulses (150 ms pulses starting from a holding potential of –80 mV with 10 mV increments) were identified as calcium (Ca$^{2+}$) currents as they were blocked by the universal voltage-activated channel inhibitor Cd$^{2+}$ (200 μM) (*McNally et al., 2020*; *Figure 2A–E*). The maximum total inward and Cd$^{2+}$-sensitive currents reached their peak amplitudes at –10 mV. To differentiate between various calcium channel subtypes present in Kiss1$^{ARH}$ neurons, we applied selective antagonists individually, allowing us to isolate the drug-sensitive current for each cell. When we individually applied specific antagonists, we observed partial inhibition of whole-cell Ca$^{2+}$ currents. Treatment with 10 μM nifedipine (*Lee et al., 2002*; *Kato et al., 2003*; *Hiraizumi et al., 2008*), an L-type Ca$^{2+}$ channel inhibitor, resulted in a partial inhibition (26.1%) of the whole-cell calcium current. Additionally, application of 2 μM $\omega$-conotoxin GVIA (conoGVIA, targeting N-type channels) (*Lee et al., 2002*), 200 nM $\omega$-agatoxin IVA (AgaIVA, targeting P/Q-type channels) (*Kato et al., 2003*; *McNally et al., 2020*), 100 nM SNX-482 (targeting R-type channels) (*Hiraizumi et al., 2008*), or 1 μM TTA-P2 (TTAP2, targeting T-type channels) (*McNally et al., 2020*) also led to partial inhibition of the Ca$^{2+}$ current (*Figure 2E*). The observed reduction in current with each inhibitor indicates the presence of all the major subtypes of Ca$^{2+}$ currents in Kiss1$^{ARH}$ neurons. Among the inhibitors used, the largest components of the whole-cell calcium current were sensitive to nifedipine (26.1%), conoGVIA (25.1%), and

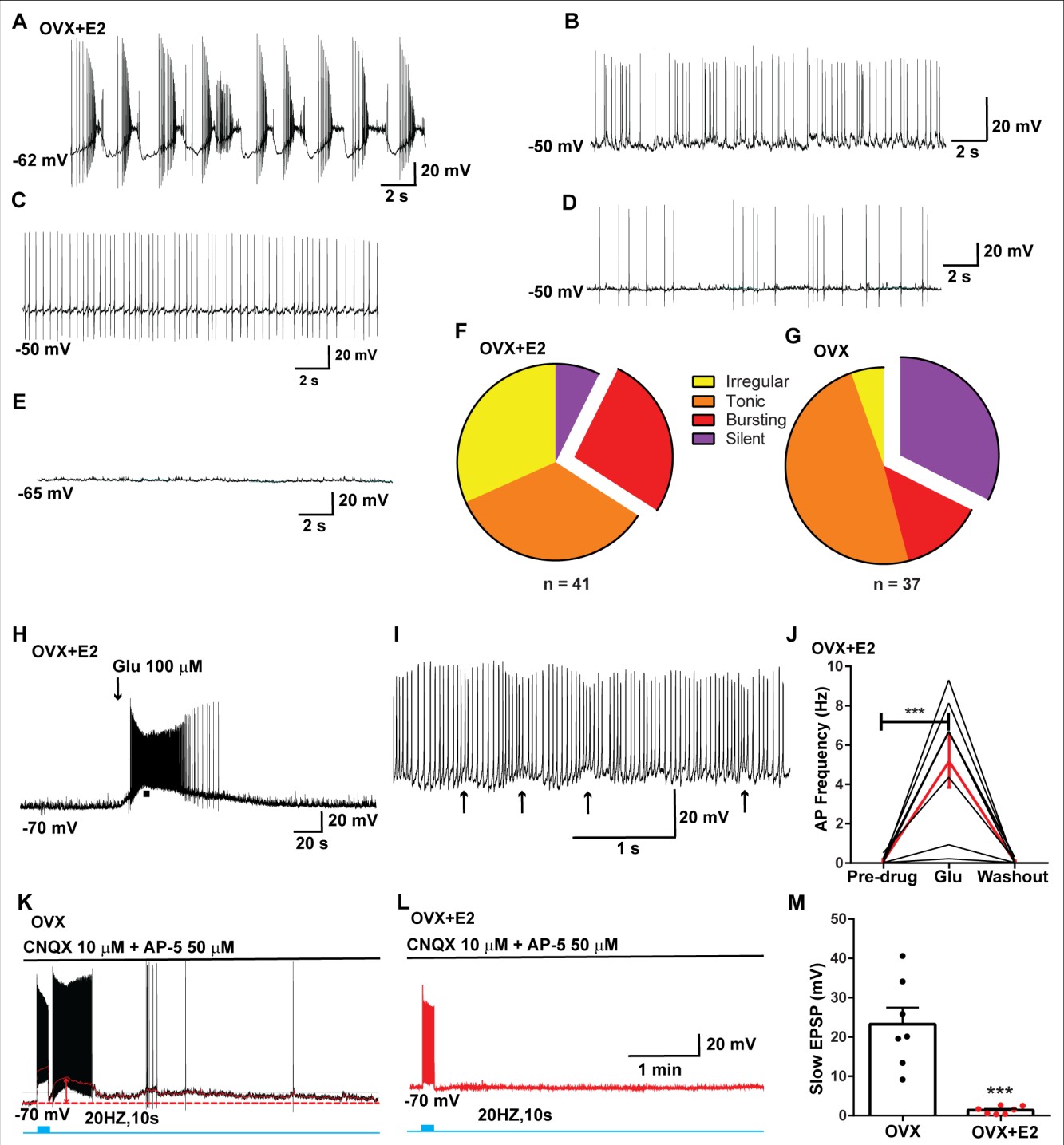

**Figure 1.** Properties of the firing of Kiss1^ARH neurons from ovariectomized (OVX) and E2-treated OVX mice. (**A–E**) Representative whole-cell, current-clamp recordings of spontaneous phasic burst firing (**A**, which is only seen in E2-treated mice with clear up and down states), irregular burst firing (**B**), tonic firing (**C**), irregular firing (**D**), and silent (**E**) in Kiss1^ARH neurons from E2-treated OVX mice. (**F, G**) Summary pie chart illustrating the distribution of firing patterns in Kiss1^ARH neurons from OVX + E2 (**F**, from six mice) or OVX (**G**, from eight mice) mice (OVX versus OVX + E2, $X^2_{(3)}$ = 6.05, p=0.0011). (**H**) Current-clamp recording in a Kiss1^ARH neuron from an OVX + E2 female demonstrating the response to glutamate (100 µM). RMP = –70 mV. Similar responses were observed in all recorded female Kiss1^ARH neurons (n = 7 from four mice). (**I**) The spiking activity above the bar in (**H**) was expanded to highlight the pronounced effects of glutamate on burst firing activity of Kiss1^ARH neurons, characterized by an ensemble of spikes riding on top of low-threshold spikes (arrows). Drugs were rapidly perfused into the bath as a 4 µl bolus. (**J**) Spontaneous AP frequency of Kiss1^ARH neurons before, during, and after glutamate application (n = 7 cells). Data are presented as mean ± SEM. Statistical comparisons between different treatments were performed

*Figure 1 continued on next page*

*Figure 1 continued*

using one-way ANOVA ($F_{(2, 18)}$ = 14.60, p=0.0002) followed by Bonferroni's post hoc test. \*\*\*p<0.005. (**K, L**) Representative traces showing the amplitude of slow excitatory postsynaptic potential (EPSP) induced by high-frequency photostimulation in Kiss1[ARH] neurons in the presence of ionotropic glutamate receptor antagonists CNQX and AP5 from OVX (**K**) and OVX + E2 (**L**) females. The arrows indicate the measurements of slow EPSP amplitude after low-pass filtering (shown in **K**). (**M**) Bar graphs summarizing the slow EPSPs in the Kiss1[ARH] neurons from OVX and E2-treated OVX mice in the presence of CNQX and AP5. Statistical comparisons between the two groups were performed using an unpaired *t*-test ($t_{(12)}$ = 5.181, p=0.0002). Data are expressed as mean ± SEM, with data points representing individual cells.

The online version of this article includes the following source data for figure 1:

**Source data 1.** Data presented in *Figure 1*.

SNX-482 (31.1%) (*Figure 2A, B, D , and F*, *Figure 2—source data 1*). Subsequently, we documented contributions from TTA-P2-sensitive channels, accounting for approximately 6.7% of the total Ca²⁺ current, and AgaIVA-sensitive channels, which constituted approximately 3.9% (*Figure 2E and C*). These findings indicate that high voltage-activated L-, N-, and R-type channels constitute the largest component of the voltage-activated Ca²⁺ current in Kiss1[ARH] neurons that not only increases the overall excitability but also greatly facilitates TRPC5 channel opening (*Blair et al., 2009*), which is the major downstream target of TACR3 activation by NKB (*Qiu et al., 2016*).

## Voltage-activated Ca²⁺ channels contribute to generation of slow EPSP in Kiss1[ARH] neurons

Our previous study utilizing optogenetics demonstrated that high-frequency photostimulation of Kiss1[ARH] neurons releases NKB. This release of NKB induces slow EPSPs that facilitates the recruitment of other Kiss1[ARH] neurons, resulting in synchronous firing of the Kiss1[ARH] neuronal population (*Qiu et al., 2016*). Additionally, chelating intracellular calcium with the fast chelator BAPTA abolishes the slow EPSP and persistent firing in Kiss1[ARH] neurons, highlighting the role of calcium signaling in these processes (*Qiu et al., 2021*). To access the involvement of high voltage-activated (HVA) Ca²⁺

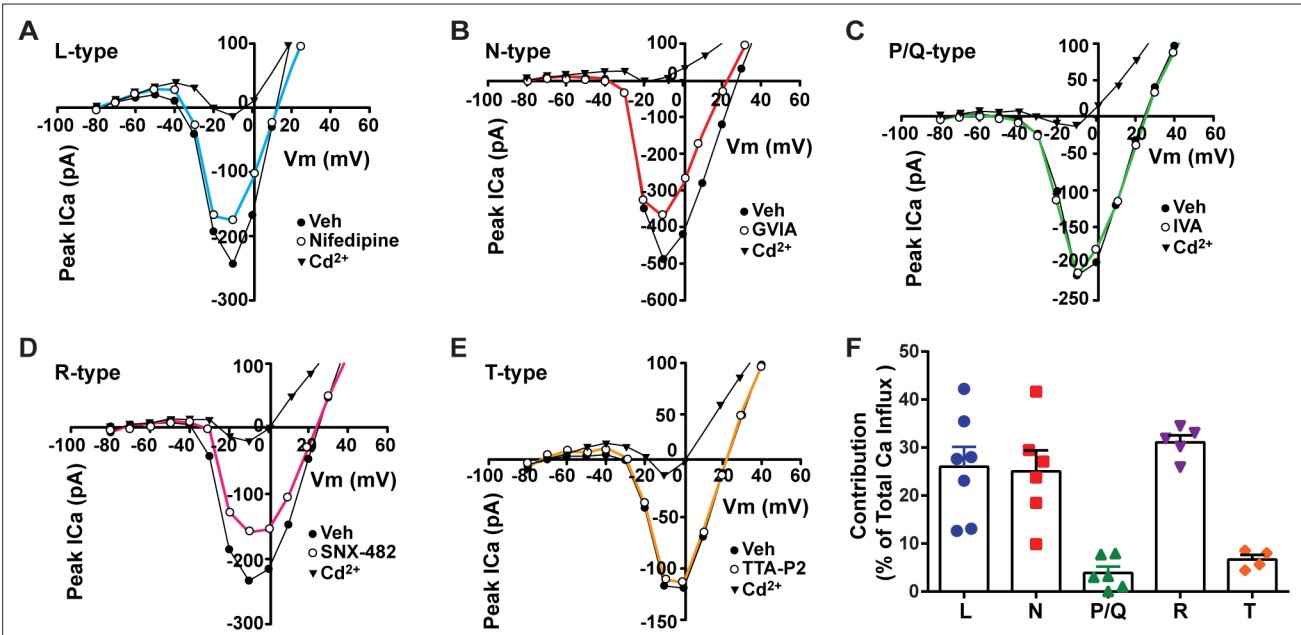

**Figure 2.** Relative contribution of voltage-gated calcium currents in *Kiss1[ARH]* neurons from OVX mice. (**A–E**) Representative current–voltage relationships showing that Cd²⁺ (non-selective blocker of calcium channels)-sensitive peak currents were inhibited by different calcium channel blockers: (**A**) nifedipine; (**B**) ω-conotoxin GIVA; (**C**) ω-agatoxin IVA; (**D**) SNX-482; (**E**) TTA-P2. (**F**) The maximum peak currents were measured at –10 mV. The proportions of Ca²⁺ currents inhibited by nifedipine (L type), ω-conotoxin GVIA (N type), ω-agatoxin IVA (P/Q), SNX-482 (R type), and TTA-P2 (T type). Data are expressed as mean ± SEM, with data points representing individual cells.

The online version of this article includes the following source data for figure 2:

**Source data 1.** Data presented in *Figure 2*.

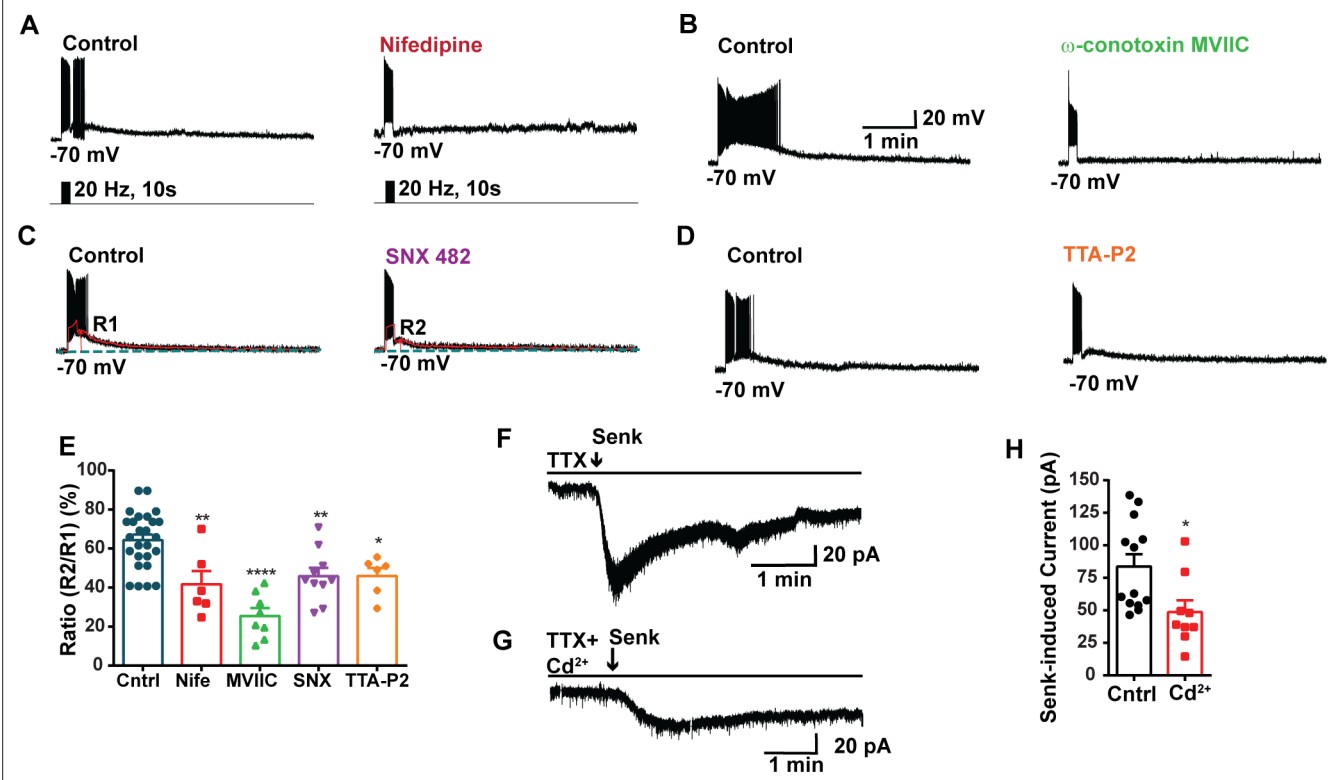

**Figure 3.** Blockade of voltage-activated Ca$^{2+}$ channels decreases the slow excitatory postsynaptic potential (EPSP) in *Kiss1$^{ARH}$* neurons. (**A–D**) Representative traces showing that the slow EPSPs induced by high-frequency photostimulation were abolished by perfusing the blocker of the L-type calcium channel, nifedipine (**A**) or N- and P/Q-type calcium channels, $\omega$-conotoxin MVIIC (**B**), or the R-type calcium channel, SNX 482 (**C**), or the T-type calcium channel, TTA-P2 (**D**), respectively. The arrows indicate the measurements of slow EPSP amplitude, denoted as R1 and R2, after low-pass filtering (shown in **C**). (**E**) Bar graphs summarizing the effects of drugs on the R2/R1 ratios. The slow EPSP was generated in OVX Kiss1-Cre::Ai32 mice. Comparisons between different treatments were performed using a one-way ANOVA (F $_{(3, 51)}$ = 14.36, p<0.0001) with the Bonferroni's post hoc test. *, **, **** indicate p<0.05, 0.01, 0.001, respectively versus control. (**F, G**) Representative traces show that in voltage clamp senktide induced a significant inward current in Kiss1$^{ARH}$ neurons (**F**) in the presence of TTX. This current was inhibited by the calcium channel blocker CdCl$_2$ (200 µM) in another cell (**G**). (**H**) Bar graphs summarize the effect of the calcium channel blocker CdCl$_2$ on the senktide-induced inward currents. Unpaired *t*-test for (**F**) versus (**G**): $t_{(20)}$ = 2.575, p=0.0181; *p<0.05. Data are expressed as mean ± SEM, with data points representing individual cells.

The online version of this article includes the following source data for figure 3:

**Source data 1.** Data presented in *Figure 3*.

channels in the generation of the slow EPSP, we conducted experiments where we blocked the L-type Ca$^{2+}$ channels with nifedipine (10 µM) and the N- and P/Q-type Ca$^{2+}$ channels with $\omega$-conotoxin MVIIC (1 µM) (*Nunemaker et al., 2003*). We then measured the slow EPSP. Indeed, both nifedipine and $\omega$-conotoxin MVIIC significantly inhibited the slow EPSP by 35.2 and 60.4%, respectively (*Figure 3A, B, and E*, *Figure 3—source data 1*). In addition, we used SNX (100 nM), which selectively blocks R-type Ca$^{2+}$ channels, and the slow EPSP was reduced to 28.7% of its control value (*Figure 3C and E*, *Figure 3—source data 1*). The selective T-channel blocker TTA-P2 (5 µM) inhibited the slow EPSP by 28.6% (*Figure 3D and E*, *Figure 3—source data 1*). Since the pharmacological blockade of the various calcium channels could also affect the presynaptic release of neurotransmitters (i.e., NKB), we measured the direct response of Kiss1$^{ARH}$ neurons to the TACR3 agonist senktide in the presence or absence of cadmium, a universal calcium channel blocker. In synaptically isolated whole-cell recordings of Kiss1$^{ARH}$ neurons, Cd$^{2+}$ significantly inhibited the inward current induced by senktide (*Figure 3F–H*, *Figure 3—source data 1*), which supports our posit that plasma membrane voltage-activated calcium channels contribute to the TRPC5 channel activation. Therefore, it appears that calcium channels contribute to maintaining the sustained depolarization underlying the slow EPSP.

## E2 increases the mRNA expression and the whole-cell current of voltage-activated Ca$^{2+}$ channels

The neuropeptides NKB (tachykinin2, *Tac2*) and kisspeptin (*Kiss1*), which are expressed in Kiss1$^{ARH}$ neurons, are crucial for the pulsatile release of gonadotropin-releasing hormone (GnRH) and reproductive processes. E2 decreases the expression of *Kiss1* and *Tac2* mRNA in Kiss1$^{ARH}$ neurons but enhances the excitability of Kiss1$^{ARH}$ neurons by amplifying the expression of *Cacna1g*, *Hcn1*, and *Hcn2* mRNA, as well as increasing T-type calcium currents and h currents, respectively (*Qiu et al., 2018*). Moreover, E2 drives *Slc17a6* mRNA expression and enhances glutamatergic synaptic input to arcuate neurons and Kiss1$^{AVPV/PeN}$ neurons (*Qiu et al., 2018*). As a result, the E2-driven increase in Kiss1$^{ARH}$ neuronal excitability and glutamate neurotransmission may play a crucial role in triggering the surge of GnRH, ultimately leading to the LH surge (*Lin et al., 2021*; *Shen et al., 2022*).

To assess the impact of E2 on the modulation of voltage-activated calcium channels and Kiss1$^{ARH}$ neuronal excitability, we employed real-time PCR (qPCR) to measure the relative expression levels of ion channel subtypes in Kiss1$^{ARH}$ neurons. We compared the expression in E2-treated females to those treated with oil using the specific primers listed in *Table 1*. The quantification was conducted on pools of 10 neurons, as indicated in the 'Materials and methods'. In both oil- and E2-treated females, we quantified the expression of *Cacna1c* (L-type), *Cacna1a* (P/Q-type), *Cacna1b* (N-type), and *Cacna1e* (R-type) mRNAs. Remarkably, all of these mRNA transcripts exhibited increased expression levels in response to E2 treatment (*Figure 4*, *Figure 4—source data 1*). Congruent with our previous findings (*Qiu et al., 2018*), we observed that the mRNA expression of *Cacna1g and Hcn1* was also upregulated in response to E2 treatment (*Figure 4*, *Figure 4—source data 1*). These results suggested that E2 has a regulatory effect on the expression and function of all of these ion channels in Kiss1$^{ARH}$ neurons. To determine whether the increased mRNA expression translated into functional changes at the cellular level, we measured the whole-cell calcium current in Kiss1$^{ARH}$ neurons obtained from ovariectomized mice treated with either vehicle or E2. We discovered that E2 treatment led to a significant increase in the peak calcium current density in Kiss1$^{ARH}$ neurons (*Figure 5A–D*, *Figure 5—source data 1*), which was recapitulated by our computational modeling (*Figure 5E*, *Figure 5—source data 1*). These findings indicate that the upregulation of the mRNA expression of calcium channels by E2 translated to an augmented peak calcium current in Kiss1$^{ARH}$ neurons.

The largest components of the calcium currents in Kiss1$^{ARH}$ neurons from the E2-treated, ovariectomized females were found to be sensitive to nifedipine, conoGVIA, and SNX-482, accounting for approximately 24.9, 24.6, and 27.0% of the total current across cells, respectively, which is very similar to their contributions to the whole-cell current from vehicle-treated, ovariectomized females (*Figure 3*). In addition, contributions from TTA-P2-sensitive channels accounted for approximately 11.1% of the total Ca$^{2+}$ current, while agaIVA-sensitive channels contributed to approximately 11.0% (*Figure 5D*, *Figure 5—source data 1*). These results highlight the prevalence of L-, N-, and R-type calcium channels as the major contributors to the whole-cell calcium current in Kiss1$^{ARH}$ neurons from E2-treated, ovariectomized females, but also the involvement of T-type and P/Q-type channels.

## E2 does not alter the kinetics of calcium channel activation or de-inactivation

In order to determine whether the increase in peak voltage-activated calcium current density was the result of E2 regulating calcium channel kinetics, mRNA expression or both, we examined the voltage dependence of activation and inactivation. By measuring the voltage dependence of activation, we assessed how E2 affects the ability of calcium channels to open in response to membrane potential changes. Similarly, by examining the voltage dependence of inactivation, we determined how E2 influences the inactivation kinetics of calcium channels. Based on our results, there was no difference in the voltage dependence of activation between Kiss1$^{ARH}$ neurons from the vehicle-treated control group ($V_{1/2}$ = −32.3 ± 2.1 mV; n = 13) and Kiss1$^{ARH}$ neurons from estrogen-treated females ($V_{1/2}$ = −33.6 ± 2.5 mV; n = 11). Similarly, there was no significant difference in the voltage dependence of inactivation between control cells ($V_{1/2}$ = −48.9 ± 4.8 mV; n = 6) and estrogen-treated cells ($V_{1/2}$ = −44.1 ± 1.9 mV; n = 5) (*Figure 6*, *Figure 6—source data 1*). Therefore, although long-term E2 treatment increased the mRNA expression of HVA calcium channels (*Figure 4*), it did not affect the channel kinetics in Kiss1$^{ARH}$ neurons (*Figure 6*). Also, E2 downregulated the expression of *Kcnd2* mRNA encoding Kv4.2, which is expressed in Kiss1$^{ARH}$ neurons (*Mendonça et al., 2018*) and has similar kinetics of activation

**Table 1.** Primer table.

| Gene name (encodes for) | Accession number | Primer location (bp) | Product length (bp) | Annealing temperature (°C) | Efficiency slope | $r^2$ | % |
|---|---|---|---|---|---|---|---|
| Cacna1c (Cav 1.2,L-type) | NM_009781 | 1331–1348 1390–1407 | 77 | 60 | –3.478 | 0.989 | 94 |
| Cacna1a (Cav 2.1,P/Q-type) | NM_007578 | 6035–6054 6090–6109 | 75 | 60 | –3.498 | 0.980 | 93 |
| Cacna1b (Cav 2.2,N-type) | NM_001042528 | 5405–5426 5467–5488 | 84 | 60 | –3.483 | 0.992 | 94 |
| Cacna1e (Cav 2.3,R-type) | NM_009782 | 530–549 617–636 | 107 | 60 | –3.441 | 0.983 | 95 |
| Cacna1g (Cav 3.1,T-type) | NM_009783 | 5004–5025 5060–5083 | 80 | 60 | –3.372 | 0.968 | 98 |
| Hcn1 (HCN1) | NM_010408 | 1527–1546 1641–1662 | 136 | 60 | –3.253 | 0.958 | 100 |
| Hcn2 (HCN2) | NM_008226 | 1122–1143 1199–1218 | 97 | 60 | –3.279 | 0.969 | 100 |
| Kcnd2 (KCND2) | NM_019697 | 2135–2156 2217–2238 | 104 | 60 | –3.312 | 0.972 | 100 |
| Kcnn3 (SK3) | NM_080466 | 1259–1277 1352–1370 | 112 | 60 | –3.369 | 0.952 | 98 |
| Kcnma1 (BKα1) | NM_001253358 | 2745–2762 2833–2852 | 108 | 60 | –3.338 | 0.971 | 99 |
| Kcnb1 (KCNB1) | NM_008420 | 691–710 759–780 | 119 | 60 | –3.375 | 0.983 | 98 |
| Kcnq2 (KCNQ2) | NM_010611 | 1079–1098 1151–1170 | 92 | 60 | –3.367 | 0.978 | 98 |
| Tac2 (NKB) | NM_009312 | 368–389 490–511 | 144 | 60 | –3.440 | 0.989 | 95 |
| Slc17a6 (VGLUT2) | NM_080853 | 1275–1296 1371–1390 | 116 | 60 | –3.374 | 0.997 | 98 |
| Trpc5 (TRPC5)* | NM_009428 | 734–753 832–851 | 118 | 60 | –3.161 | 0.952 | 100 |
| Trpc5 (TRPC5)† | NM_009428 | 616–637 772–792 | 177 | 60 | –3.407 | 0.987 | 97 |
| Trpc5 (TRPC5)† | NM_009428 | 2083–2100 2162–2179 | 97 | 60 | –3.255 | 0.963 | 100 |
| Kcnj6 (GIRK2) | NM_001025584 | 488–507 592–609 | 122 | 60 | –3.368 | 0.919 | 98 |
| Gapdh (GAPDH) | NM_008084 | 689–706 764–781 | 93 | 60 | –3.352 | 0.998 | 99 |

*Trpc5 primers (**Figure 10E**).

†Trpc5 primers flanking sgRNA and PAM sites in mutagenesis (**Figure 11D**).

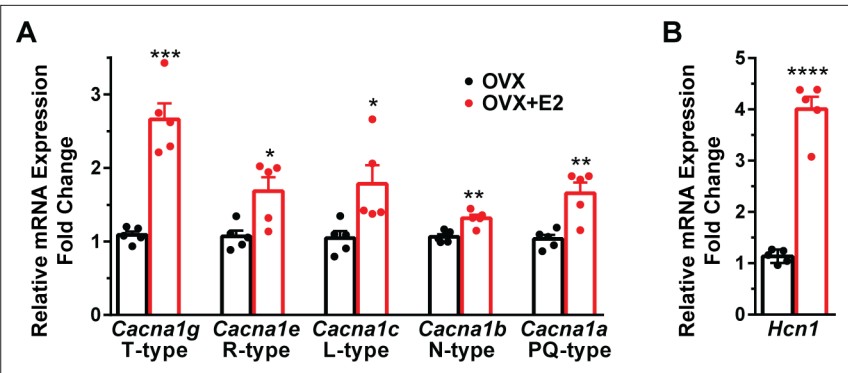

**Figure 4.** E2 increases the mRNA expression of volatge-activated Ca²⁺ channels and *Hcn1* channels in Kiss1^ARH neurons. (**A**) E2 increases the expression of low and high voltage-activated calcium channels in Kiss1^ARH neurons. Kiss1^ARH neurons (three 10-cell pools) were harvested from each of five vehicle- and five E2-treated, OVX females to quantify ion channel mRNA expression of low and high voltage-activated calcium channels as described in the 'Materials and methods'. The analysis included T-type (*Cacna1g*) low voltage-activated, as well as the following high voltage-activated channels: R-type (*Cacna1e*), L-type (*Cacna1c*), N-type (*Cacna1b*), and P/Q-type (*Cacna1a*) calcium channels. Interestingly, all of these channels were upregulated with E2 treatment, which significantly increased the whole-cell calcium current (see **Figure 5**). Bar graphs represent the mean ± SEM, with data points representing individual animals (oil versus E2, unpaired *t*-test for *Cacna1g*, $t_{(8)}$ = 7.105, ***p=0.0001; for *Cacna1e*, $t_{(8)}$ = 3.007, *p=0.0169; for *Cacna1c*, $t_{(8)}$ = 2.721, *p=0.0262; for *Cacna1b*, $t_{(8)}$ = 4.001, **p=0.0039; for *Cacna1a*, $t_{(8)}$ = 4.225, **p=0.0028). (**B**) The same Kiss1^ARH neuronal pools were also analyzed for mRNA expression of *Hcn1* ion channels, and E2 also increased the expression of hyperpolarization-activated, cyclic-nucleotide gated *Hcn1* channels in Kiss1^ARH neurons. *Hcn1* channel mRNA expression was the most highly upregulated by E2 treatment in Kiss1^ARH neurons. The expression values were calculated via the ΔΔCT method, normalized to *Gapdh* and relative to the oil control values (oil versus E2, unpaired *t*-test, $t_{(8)}$ = 11.450, ****p<0.0001).

The online version of this article includes the following source data for figure 4:

**Source data 1.** Data presented in **Figure 4**.

as the T-type calcium channels (oil-treated, ovariectomized females relative mRNA expression: 1.053, n = 5 animals versus E2-treated expression: 0.5643, n = 5; *t*-test p=0.0061). This opposing K⁺ current would dampen the inward calcium current. Therefore, it appears that long-term E2 treatment does not modulate calcium channel kinetics but rather alters the mRNA expression to increase the calcium channel conductance.

## BK, SK, and KCNQ channels are involved in modulating excitability of Kiss1^ARH neurons

In the brain, calcium plays a crucial role in sculpting neuronal firing by activating potassium channels, which subsequently influence neuronal behavior (*Nicoll, 1988*; *Storm, 1990*). Since HVA and LVA calcium channels were expressed in Kiss1^ARH neurons, all of which contribute to the elevation of intracellular calcium concentration ([Ca²⁺]ᵢ) that facilitates TRPC5 channel opening (*Blair et al., 2009*), our next step involved measuring the changes in Ca²⁺-activated K⁺ channel conductances and assessing their mRNA expression. In various cell types, increases in cytosolic calcium levels, whether resulting from extracellular influx or intracellular release, lead to the activation of plasma membrane calcium-activated potassium channels (*Sah and Faber, 2002*). Similarly, in Kiss1^ARH neurons, these channels would be activated by calcium influx through all four types of HVA calcium channels, as well as the LVA calcium channel, which are all active during action potential firing. The activity of Ca²⁺-activated K⁺ channels plays a crucial role in numerous physiological processes, including secretion and the regulation of neuronal firing properties. Two main families of Ca²⁺-activated K⁺ channels have been characterized, distinguished by their biophysical and pharmacological properties. These families are known as BK (big conductance K⁺) and SK (small conductance K⁺) channels in the CNS (*Kshatri et al., 2018*). BK channels are known for their high potassium selectivity and large single-channel conductance, typically ranging from 100 to 300 pS. Activation of BK channels requires both calcium binding and membrane depolarization (*Blatz and Magleby, 1987*; *Marty, 1989*; *Storm, 1990*; *Sah, 1996*).

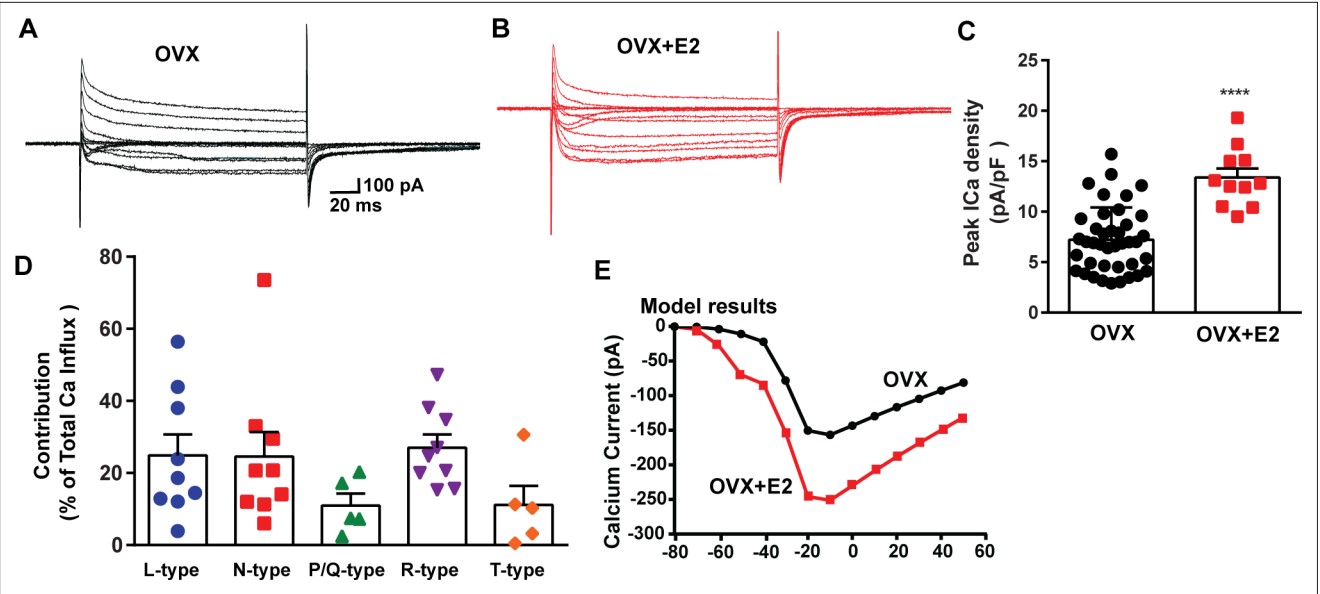

**Figure 5.** E2 treatment (positive feedback regimen) increases the $Ca^{2+}$ currents in $Kiss1^{ARH}$ neurons. (**A, B**) $Ca^{2+}$ currents in $Kiss1^{ARH}$ neurons with the same membrane capacitance from oil-treated (**A**) or E2-treated (**B**) animals. (**C**) The maximum peak currents were measured at –10 mV. The current amplitudes were normalized to the cell capacitance in all cases to calculate current density. The bar graphs summarize the density of $Ca^{2+}$ current in $Kiss1^{ARH}$ neurons from oil-treated and E2-treated animals. The mean density was significantly greater in E2-treated (13.4 ± 0.9 pA/pF, n = 11) than in oil-treated OVX females (7.2 ± 0.5 pA/pF, n = 40) (unpaired t-test, $t_{(49)}$ = 5.75, ****p<0.0001). (**D**) Relative contribution of voltage-gated calcium currents in $Kiss1^{ARH}$ neurons from OVX, E2-treated mice. The maximum peak currents were measured at –10 mV. The proportions of $Ca^{2+}$ currents inhibited by nifedipine (L type), $\omega$-conotoxin GVIA (N type), $\omega$-agatoxin IVA (P/Q), SNX-482 (R type), and TTA-P2 (T type). Data are expressed as mean ± SEM, with data points representing individual cells. (**E**) The modeling predicts that E2-treated, OVX females exhibit a significantly greater inward $Ca^{2+}$ current (red trace) than the vehicle-treated females (black trace). The conductance of the modeled voltage-gated calcium current (L-, N-, P/Q-, and R-type) is set to 2.1 nS in the OVX state and 2.8 nS in OVX + E2 state, while for the T-type is set to 0.66 nS in the OVX state and 5 nS in OVX + E2 state.

The online version of this article includes the following source data for figure 5:

**Source data 1.** Data presented in *Figure 5*.

---

On the other hand, SK channels are simply activated by increases in cytosolic calcium levels, with their half-maximal activation at 0.3 µM (***Bond et al., 1999***).

To investigate $K^{+}$ currents, the cells were maintained at a holding potential of –70 mV while being exposed to the blockers CNQX, AP5, picrotoxin, and TTX. Subsequently, the membrane potential was stepped by depolarizing voltages, ranging from –60 mV to +40 mV in 10 mV increments, for a duration of 500 ms (***Brereton et al., 2013***). This protocol was employed to activate $K^{+}$ currents (***Figure 7A***). First, we examined SK currents. The mean current density was determined at the end of the voltage pulses. In vehicle-treated, OVX females, the application of the SK channel blocker apamin (100 nM) (***Spergel, 2007***) led to a significant reduction in whole-cell currents in the +10 to +40 mV range (***Figure 7A and B***, ***Figure 7—source data 1***). The mean outward current density at +40 mV in the control group was 125.5 ± 7.2 pA/pF (n = 5) with the apamin-sensitive component contributing 60.2 ± 3.3 pA/pF (n = 5) (***Figure 7A and C***). In contrast, in the E2-treated females, the overall mean outward current density at +40 mV was 195.4 ± 14.6 pA/pF (n = 6), which was significantly greater than the vehicle control group (***Figure 7D and E***, ***Figure 7—source data 1***). However, there was no significant difference in the apamin-sensitive component between the vehicle-treated and E2-treated females, 60.2 ± 3.3 pA/pF (n = 5) versus 75.9 ± 12.3 pA/pF (n = 6), respectively (***Figure 7F***, ***Figure 7—source data 1***). Furthermore, to investigate the expression of the mRNAs encoding SK channel subunits in Kiss1$^{ARH}$ neurons from vehicle-treated and E2-treated OVX females, qPCR experiments were performed on 10-cell Kiss1$^{ARH}$ neuronal pools (***Figure 7G***, ***Figure 7—source data 1***). We focused on SK3 channels because these channels exhibit the highest expression in the hypothalamus and E2 regulates their expression (***Bosch et al., 2002***). E2 treatment had no effect on the mRNA expression of the *Kcnn3* (SK3) subunit. These molecular findings support our electrophysiology results. Our

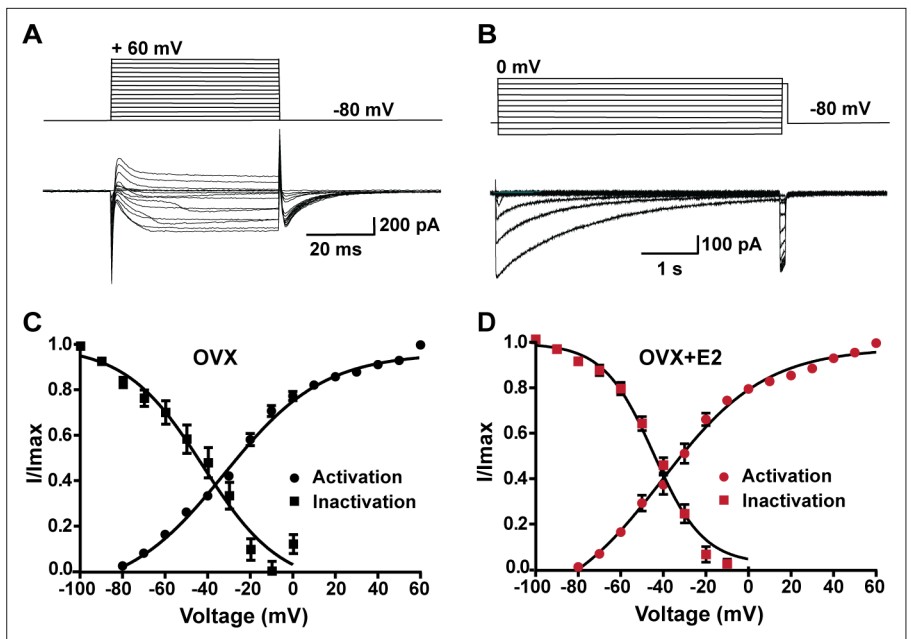

**Figure 6.** Voltage dependence of $I_{Ca}$ in *Kiss1^ARH* neurons from OVX and OVX + E2 mice. (**A, B**) Top panels: activation and inactivation protocol. Bottom: representative traces. (**C, D**) The mean $V_{1/2}$ values for calcium channel activation were not significantly different for cells from controls versus cells from E2-treated females. Similarly, the $V_{1/2}$ values for channel steady-state inactivation were similar for both groups. Data are expressed as mean ± SEM.

The online version of this article includes the following source data for figure 6:

**Source data 1.** Data presented in *Figure 6*.

computational model was calibrated so that SK channels contributed ~50 pA/pF to the whole-cell outward K⁺ current in E2-treated females (*Figure 7H*, *Figure 7—source data 1*).

Additionally, following the same protocol and in the presence of the same cocktail of blockers (CNQX, AP5, picrotoxin, and TTX), we investigated the contribution of BK channels to Kiss1^ARH neuronal excitability. In the OVX females, the application of the BK channel blocker iberiotoxin (ibTx, 200 nM) (*Niday and Bean, 2021*) resulted in only a slight attenuation of the outward current (n = 5) (*Figure 8A and B*, *Figure 8—source data 1*). The ibTx-sensitive current density measured at +40 mV was 32.8 ± 12.0 pA/pF (*Figure 8A and C*, *Figure 8—source data 1*). However, in the E2-treated females, the application of ibTx significantly attenuated the whole-cell K⁺ current from +20 to +40 mV, (*Figure 8D and E*, *Figure 8—source data 1*). Additionally, the ibTx-sensitive current was significantly larger in the +0 to +40 mV range in the E2-treated females compared to the OVX females (*Figure 8F*, *Figure 8—source data 1*). These findings indicate that E2 treatment modulates the activity of ibTx-sensitive BK current in Kiss1^ARH neurons, resulting in increased current density (90.0 ± 10.1 pA/pF [n = 6] versus 31.1 ± 8.4 pA/pF [n = 5] at +40 mV). To investigate the expression of mRNA encoding BK channel subunits in Kiss1^ARH neurons from vehicle-treated and E2-treated OVX females, qPCR experiments were performed on 10-cell Kiss1^ARH neuronal pools (*Figure 8G*, *Figure 8—source data 1*). E2 treatment significantly increased the mRNA expression of the *Kcnma1* (BKα1) subunit. These findings support our electrophysiological findings that there is a significant increase in BK channel activity in Kiss1^ARH neurons with E2 treatment (*Figure 8F*). In addition, E2 increased the mRNA expression of *Kcnb1* encoding delayed rectifier Kv2.1 (E2-treated relative mRNA expression: 1.672, n = 5 versus oil-treated mRNA expression: 1.086, n = 5; *t*-test p-value=0.0024). The upregulation of two of these K⁺ channels (BKα1 and Kv2.1) would facilitate rapid repolarization of Kiss1^ARH following an action potential. Our computational model was calibrated so that BK channels contributed ~100 pA/pF to the whole-cell outward K⁺ current in the E2-treated females (*Figure 8H*, *Figure 8—source data 1*).

Traditionally, the afterhyperpolarization is divided into three distinct phases: fast (fAHP), medium (mAHP), and slow afterhyperpolarization (sAHP) (*Storm, 1990*; *Vogalis et al., 2003*). The fAHP is primarily mediated by the BK family of potassium channels (*Storm, 1987*). The mAHP is predominantly

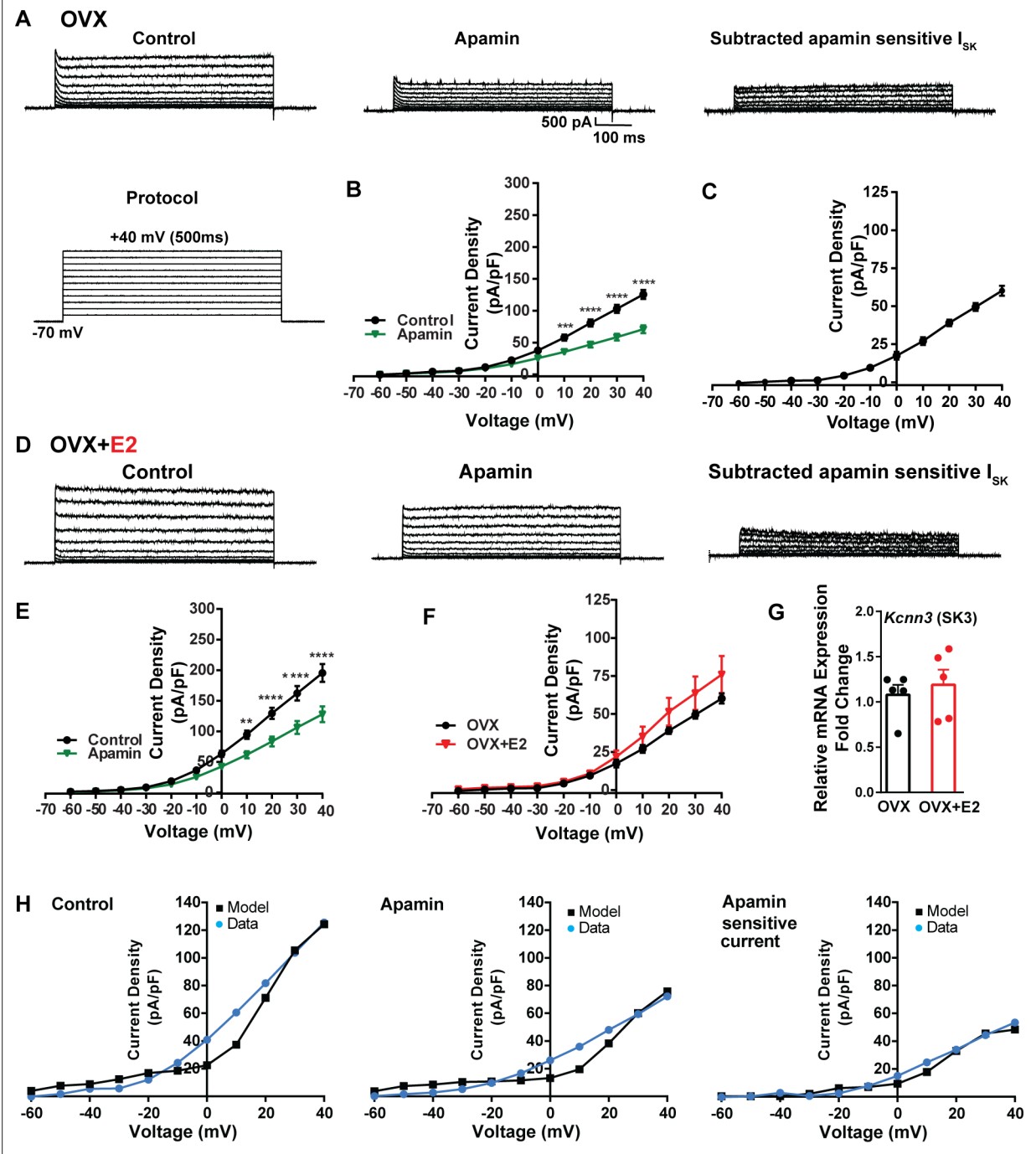

**Figure 7.** Contribution of small conductance, calcium-activated K+ (SK) channel to *Kiss1ARH* neuronal excitability. (**A**) Representative traces of the inhibition of outward currents before (left, control) and after the specific SK blocker apamin (500 nM, middle). Apamin-sensitive currents were calculated from the subtraction of control and apamin at depolarized potentials (right). Cells were clamped at –70 mV and given 500 ms voltage pulses from –60 mV to +40 mV in 10 mV steps at 0.2 Hz, as shown in (**A**) at the bottom. (**B**) Mean current density–voltage relationships measured at the end of the 500 ms voltage step ranging from –60 mV to +40 mV were obtained in the absence and presence of apamin (two-way ANOVA: main effect of treatment [$F_{(1, 8)}$ = 22.69, p=0.0014], main effect of voltage [$F_{(10, 80)}$ = 306.0, p<0.0001] and interaction [$F_{(10, 80)}$ = 24.76, p<0.0001]; mean ± SEM; n = 5; post hoc Bonferroni test, ***p<0.005, ****p<0.001). (**C**) Apamin-sensitive current densities were obtained from (**B**) (mean ± SEM, n = 5). (**D**) Representative traces of the inhibition of outward currents before (left, control) and after the specific SK blocker apamin (500 nM, middle). Apamin-sensitive currents resulted from the subtraction of control and apamin at depolarized potentials (right). (**E**) Mean current density–voltage relationships measured at the end of the 500 ms voltage step ranging from –60 mV to +40 mV were obtained in the absence and presence of apamin (two-way ANOVA: main effect of treatment [$F_{(1, 10)}$ = 12.85, p=0.0050], main effect of voltage [$F_{(10,100)}$ = 264.1, p<0.0001] and interaction [$F_{(10, 100)}$=11.93, p<0.0001]; mean ± SEM, n = 6;

*Figure 7 continued on next page*

*Figure 7 continued*

post hoc Bonferroni test, **p<0.01, ****p<0.001). (**F**) Apamin-sensitive current densities were obtained from (**C**) and (**E**) (ns; two-way ANOVA followed by Bonferroni post hoc test; mean ± SEM; OVX, n = 5; OVX + E2, n = 6). (**G**) Kiss1[ARH] neurons (three 10-cell pools) were harvested from each of five vehicle- and five E2-treated, OVX females to quantify the mRNA expression of *Kcnn3* ion channel. E2 did not increase the mRNA expression of small conductance calcium-activated K[+] (SK3) channels in Kiss1[ARH]. Bar graphs represent the mean ± SEM, with data points representing individual animals, oil versus E2, Unpaired *t*-test, $t_{(8)}$ = 0.551, p=0.5967. The expression values were calculated via the ΔΔCT method, normalized to *Gapdh* and relative to the oil control values. (**H**) The mathematical model was calibrated on the electrophysiology data from Kiss1[ARH] neurons before and after treatment with the specific SK blocker apamin, left panel versus middle panel, respectively. For the calibration, it was assumed that the applied concentration of apamin (500 nM) completely blocked the SK current. The modeled apamin-sensitive current with $g_{SK}$ = 28.1 nS (right panel) matches the electrophysiological data from OVX animals.

The online version of this article includes the following source data for figure 7:

**Source data 1.** Data presented in *Figure 7*.

mediated by apamin-sensitive SK2 channels (*Bond et al., 2004*; *Peters et al., 2005*). However, KCNQ family members contribute to both the mAHP and sAHP (*Tzingounis and Nicoll, 2008*; *Tzingounis et al., 2010*). Therefore, to investigate the contribution of KCNQ channels to Kiss1[ARH] neuronal excitability, voltage-clamp experiments were conducted in the presence of TTX, CNQX, AP5, and picrotoxin, and a standard M current protocol was run using the M-channel blocker XE-991 to isolate the M current (*Roepke et al., 2011*; *Greene et al., 2017*; *Figure 9A and B*). The application of XE-991 resulted in the inhibition of M current within a physiologically relevant voltage range of –60 to –30 mV in E2-treated OVX females but exhibited minimal impact in OVX females (*Figure 9C and D*, *Figure 9—source data 1*). Although the XE991-sensitive current was relatively small compared to other voltage-activated K[+] conductances, it demonstrated a significant increase in E2-treated, OVX females (*Figure 9E*, *Figure 9—source data 1*). The maximum peak current density sensitive to XE-991 at –30 mV was found to be four times higher in E2-treated OVX females compared to OVX females. This would contribute to the repolarization following burst firing. Furthermore, E2 increased the mRNA expression of *Kcnq2* (*Figure 9F*, *Figure 9—source data 1*), which suggests that KCNQ channels play a role in repolarizing Kiss1[ARH] neurons following burst firing. Indeed, our modeling predicted that M current contributed to the repolarization following burst firing (*Figure 9G*).

## E2 increases *Slc17a6* but downregulates *Tac2*, *Trpc5*, and *Kcnj6* mRNA expression in Kiss1[ARH] neurons

Based on our electrophysiological results, Kiss1[ARH] neurons appear to transition from peptidergic to glutamatergic neurotransmission through E2-mediated changes in the expression of voltage-activated $Ca^{2+}$ channels and K[+] channels and their respective conductances. Therefore, we asked the question is there a difference in peptide and glutamate mRNA expression mediating this transition? Therefore, we ran a comparison between *Tac2* (NKB) and *Slc17a6* (surrogate for glutamate) expression. The cycle threshold (CT) was compared between *Tac2* and *Slc17a6* as well as *Kiss1*, *Trpc5*, and *Kcnj6* in Kiss1[ARH] neuronal cell pools from OVX oil-treated and OVX E2-treated animals (*Figure 10A and B*). It is worth noting that lower number of cycles illustrate a higher quantity of mRNA expression because the fluorescence is detected earlier, and one cycle difference represents a doubling in expression. As expected, the reference gene *Gapdh* did not change with E2 treatment. However, quantitative PCR results revealed that E2 treatment of OVX females significantly reduced *Tac2* expression (*Figure 10C*, *Figure 10—source data 1*), whereas *Slc17a6* mRNA was significantly increased in Kiss1[ARH] neurons (*Figure 10D*, *Figure 10—source data 1*). Moreover, both *Trpc5* and *Kcnj6* expression were significantly reduced in E2-treated, OVX females (*Figure 10E and F*, *Figure 10—source data 1*).

## CRISPR mutagenesis of *Trpc5* attenuates slow EPSP and reduces excitability of Kiss1[ARH] neurons

Our previous (*Qiu et al., 2021*) and current findings suggested that TRPC5 channels play a dominant role in regulating cell excitability in ovariectomized females. Therefore, as proof of principle, we utilized a CRISPR approach to mutate *Trpc5* channels in Kiss1[ARH] neurons similar to our previous studies (*Hunker et al., 2020*; *Stincic et al., 2021*). Hunker et al. developed a single viral vector for conditional expression of the smaller *Staphylococcus aureus* (SaCas9) and sgRNA that yields high-efficiency mutagenesis in specific cell types (*Hunker et al., 2020*). To selectively mutate *Trpc5* in

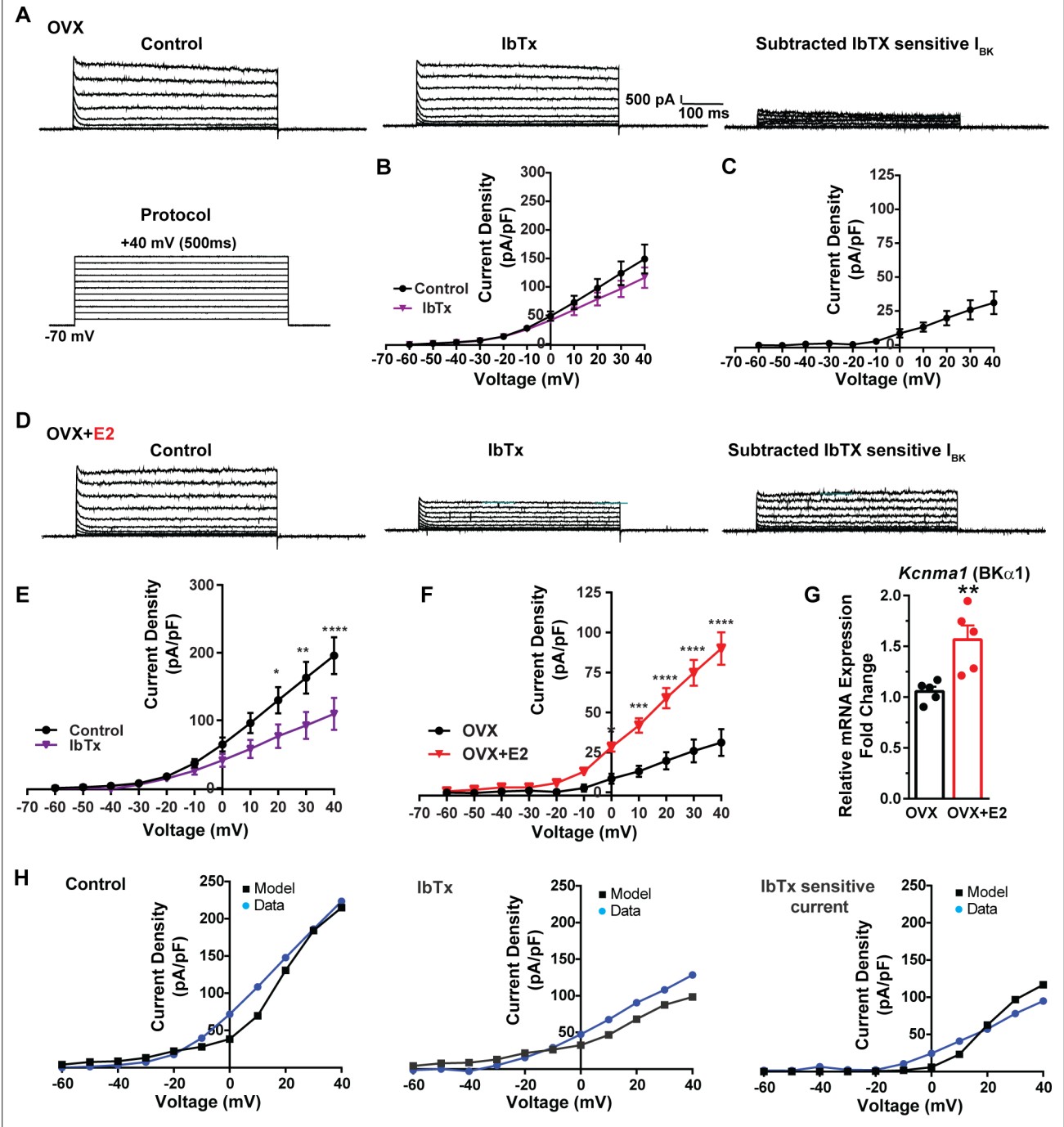

**Figure 8.** E2 upregulates *Kcnma1* mRNA and BK current in Kiss1[ARH] neurons. (**A**) Representative traces of the inhibition of outward currents before (left, control) and after the specific BK blocker iberiotoxin (IbTx; 200 nM, middle). IbTx-sensitive currents were calculated from the subtraction of control and IbTx at depolarized potentials (right). Cells were clamped at –70 mV and given 500 ms voltage pulses from –60 mV to +40 mV in 10 mV steps at 0.2 Hz, as shown in (**A**) at the bottom. (**B**) Mean current density–voltage relationships measured at the end of the 500 ms voltage step ranging from –60 mV to +40 mV were obtained in the absence and presence of IbTx (two-way ANOVA: main effect of treatment [$F_{(1, 8)}$ = 0.8841, p=0.3746], main effect of voltage [$F_{(10, 80)}$ = 71.56, p<0.0001] and interaction [$F_{(10, 80)}$ = 1.127, p=0.3528]; mean ± SEM, n = 5; post hoc Bonferroni test, p>0.05). (**C**) IbTX-sensitive current densities were obtained from (**B**) (mean ± SEM, n = 5). (**D**) Representative traces of the inhibition of outward currents before (left, control) and after the specific BK blocker iberiotoxin (IbTx; 200 nM, middle). IbTx-sensitive currents resulted from the subtraction of control and IbTx at depolarized potentials (right). (**E**) Mean current density–voltage relationships measured at the end of the 500 ms voltage step ranging from –60 mV to +40 mV were obtained in the absence and presence of IbTX (two-way ANOVA: main effect of treatment [$F_{(1, 10)}$ = 4.660, p=0.0562], main effect of voltage [$F_{(10, 100)}$ = 63.98, p<0.0001] and interaction [$F_{(10, 100)}$ = 4.907, p<0.0001]; mean ± SEM, n = 6; post hoc Bonferroni test, *p<0.05, **p<0.01, ****p<0.001). (**F**) IbTx-

*Figure 8 continued on next page*

*Figure 8 continued*

sensitive current densities were obtained from (**C**) and (**E**) (two-way ANOVA: main effect of treatment [$F_{(1, 9)}$ = 22.04, p=0.0011], main effect of voltage [$F_{(10, 90)}$ = 78.26, p<0.0001] and interaction [$F_{(10, 90)}$ = 17.84, p<0.0001]; mean ± SEM, OVX, n = 5; OVX + E2, n = 6; Bonferroni post hoc test, *p<0.05, ***p<0.005, ****p<0.001). (**G**) Kiss1[ARH] neurons (three 10-cell pools) were harvested from each of five vehicle- and five E2-treated, OVX females to quantify the mRNA expression of *Kcnma1* (BKα1) channel. E2-treatment increased the mRNA expression of *Kcnma1*. The expression values were calculated via the ΔΔCT method, normalized to *Gapdh* and relative to the oil control values. Bar graphs represent the mean ± SEM, with data points representing individual animals (unpaired two-tailed *t*-test for BK, $t_{(8)}$ = 3.479, **p=0.0083). (**H**) The mathematical model was calibrated to reproduce the current–voltage relationship observed in Kiss1[ARH] neurons before and after treatment with IbTx. For the calibration, it was assumed that the applied concentration of IbTx (200 nM) completely blocked the BK current. The modeled IbTx -sensitive current with $g_{BK}$ = 20.0 nS (right panel) matches the electrophysiological data from OVX + E2 animals.

The online version of this article includes the following source data for figure 8:

**Source data 1.** Data presented in *Figure 8*.

Kiss1[ARH] neurons, we generated two guide RNAs, one targeting exon 2, which is conserved across all splice variants, and the other targeting exon 7, the pore forming domain (*Figure 11A and B*). A cohort of Kiss1[ARH] mice were given bilateral stereotaxic injections into the ARH of the two AAV1-FLEX-SaCas9-sg*Trpc5*'s or a control virus containing with AAV1-FLEX-SaCas9-U6-sg*Rosa26*. An additional Cre-dependent virus of the same serotype (AAV1) that drove expression of a fluorophore (YFP or mCherry) was co-administered in order to visualize injection quality and facilitate harvesting of cells (*Figure 11C*). After 3 weeks, mice underwent ovariectomy since OVX mice express the maximum slow EPSP amplitude (*Qiu et al., 2016*). Brain slices were prepared, and cells harvested as previously described (*Qiu et al., 2018*) and analyzed with qPCR. We found that the *Trpc5* mutagenesis group displayed a reduction in the relative expression of *Trpc5* in Kiss1[ARH] neurons compared to the control group (*Figure 11D*, *Figure 11—source data 1*). Hence, the qPCR data verified that in the sg*Trpc5*-targeted mice we can selectively reduce *Trpc5* gene expression in targeted cells.

As predicted, mutagenesis of *Trpc5* channels in Kiss1[ARH] neurons significantly attenuated the slow EPSP (*Figure 12A–C*, *Figure 12—source data 1*) such that the postsynaptic excitation was reduced to a 'trickle' of action potential firing. What we would not have predicted is that the double sgRNA mutagenesis of *Trpc5* channels in Kiss1[ARH] neurons significantly hyperpolarized the resting membrane potential by 7 mV (*Figure 12D*, *Figure 12—source data 1*). Moreover, the rheobase (minimum current required to induce firing) significantly increased by ~13% in females bearing the *Trpc5* double mutagenesis (*Figure 12E*, *Figure 12—source data 1*). The firing frequency versus injected current (F-I) curve for *Trpc5* double mutagenesis Kiss1[ARH] neurons was also significantly attenuated (*Figure 12F*, *Figure 12—source data 1*). It would be of interest in future experiments to do the reciprocal experiment to see if overexpressing *Trpc5* channels in Kiss1[ARH] neurons from OVX + E2 females restores the RMP and 'rescues' the synchronization phenotype. However, in agreement with the present experimental findings, current ramp simulations of our mathematical model confirmed that a decrease in the TPRC5 conductance increased the rheobase and reduce the firing activity of Kiss1[ARH] neurons (*Figure 12G and H*, *Figure 12—source data 1*). In our model, this contribution of TRPC5 channels to cell excitability is attributed to intracellular calcium levels that activate TRPC5 channels independently of NKB stimulation. Finally, we employed our mathematical model to further investigate the transition from synchronous firing driven by NKB release and TRPC5 channel activation to burst firing generated by E2-mediated upregulation of endogenous conductances. As we varied the conductance of TRCP5 channels and GIRK channels under saturating extracellular signals (e.g., NKB and Dyn), we find that balance between the two conductances controls the neuronal excitability in the OVX-parameterized model, with synchronous firing eliminated when TRPC5 conductance is low (red triangle) relative to the GIRK conductance (*Figure 13A*, *Figure 13—source data 1*). Furthermore, the burst firing of the OVX + E2 parameterized model was supported by elevated h- and Ca²⁺ currents (*Figure 13B*, *Figure 13—source data 1*) as well as by the high conductance of voltage-activated Ca²⁺ channels relative to the conductance of TRPC5 channels (*Figure 13C*, *Figure 13—source data 1*). Importantly, the simulated burst firing was analogous to the phasic burst firing recorded ex vivo in Kiss1[ARH] neurons (*Figure 1A*).

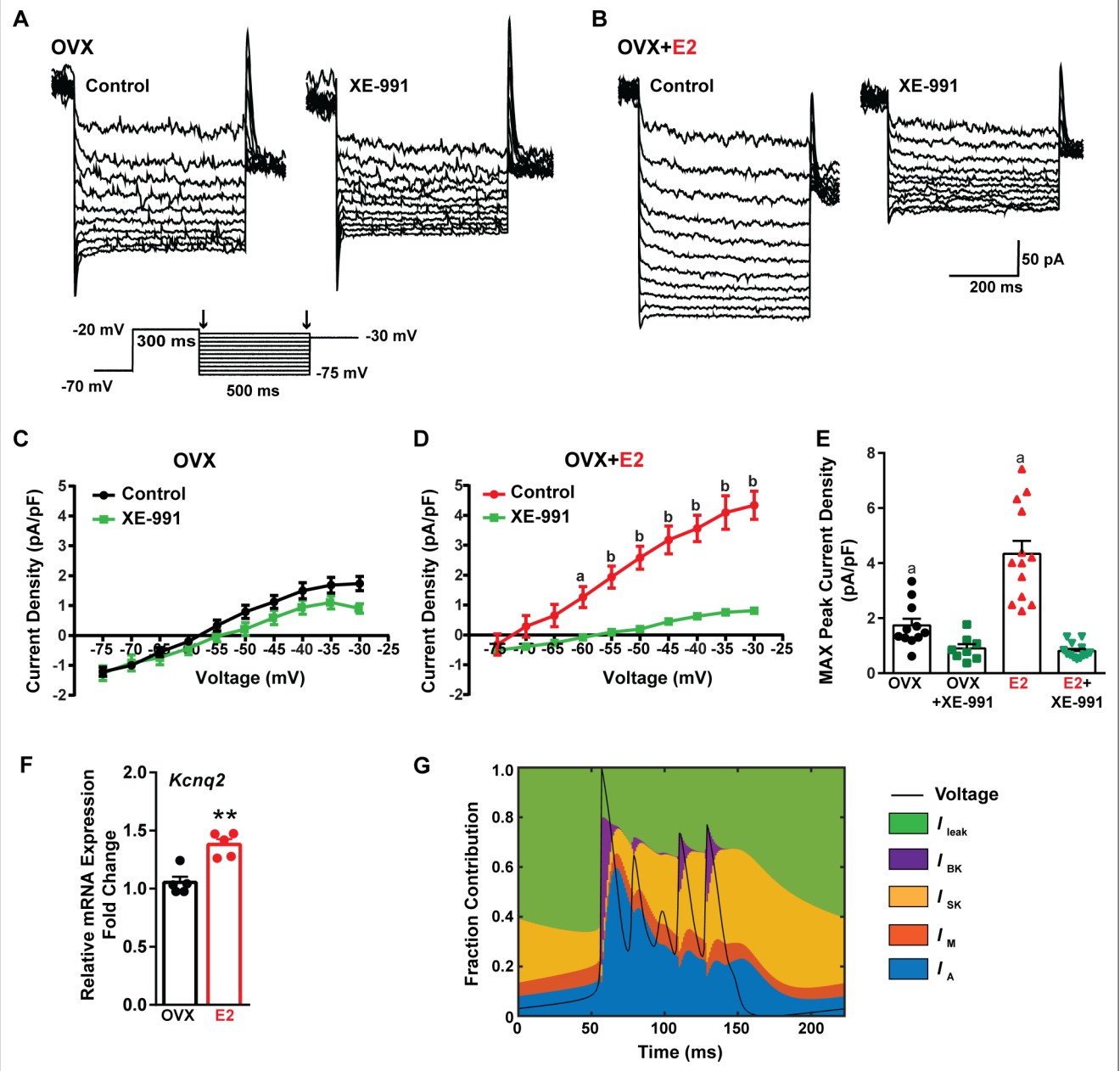

**Figure 9.** E2 upregulates *Kcnq2* channels and M current in *Kiss1*[ARH] neurons. (**A, B**) Representative current traces of the M current inhibition caused by 40 μM XE-991 perfused for 10 min in (**A**) OVX oil and (**B**) OVX + E2-treated female mice. Inset: M current deactivation protocol. (**C, D**) Current density–voltage plots from –75 to –30 mV of vehicle and XE-991 perfusion in (**C**) OVX oil and (**D**) OVX + E2-treated mice. Two-way ANOVA for (**C**): main effect of treatment ($F_{(1, 17)}$ = 1.908, p=0.1851), main effect of voltage ($F_{(9, 153)}$ = 187.1, p<0.0001) and interaction ($F_{(9, 153)}$ = 3.901, p=0.0002); Veh, n = 11; XE-991, n = 8; Bonferroni post hoc test, p>0.05. For (**D**): main effect of Veh and XE-991 ($F_{(1, 24)}$ = 24.92, p<0.0001), main effect of voltage ($F_{(9, 216)}$ = 174.5, p<0.0001) and interaction ($F_{(9, 216)}$ = 52.75, p<0.0001); Veh, n = 13; XE-991, n = 13; Bonferroni post hoc test, a = p<0.05, b = p<0.001. (**E**) Treatment with E2 elevated, while XE-991 diminished the maximum peak current density elicited by a –30 mV step in OVX- and OVX + E2-treated mice. Two-way ANOVA: main effect of Veh and XE-991 ($F_{(1, 41)}$ = 47.59, p<0.0001), main effect of OVX and OVX + E2 ($F_{(1, 41)}$ = 15.76, p=0.0003), and interaction $F_{(1, 41)}$ = 18.2, p=0.0001; Bonferroni post hoc test, Veh: OVX vs. OVX + E2, a = p<0.001. XE-991: OVX vs. OVX + E2, p>0.05. Data are expressed as mean ± SEM, with data points representing individual cells. (**F**) Kiss1[ARH] neurons (three 10-cell pools) were harvested from each of five vehicle- and five E2-treated, OVX females to quantify the mRNA expression of *Kcnq2*. E2 treatment increased the mRNA expression of *Kcnq2*. Unpaired *t*-test, $t_{(8)}$ = 4.850, **p=0.0013. Data are expressed as mean ± SEM, with data points representing individual animals. (**G**) Percent contribution of the different K⁺ currents to the repolarization current during burst-type firing activity in the OVX + E2 state. At each time point, the length of each color bar denotes the percent contribution of the corresponding current to the total outward current.

The online version of this article includes the following source data for figure 9:

**Source data 1.** Data presented in *Figure 9*.

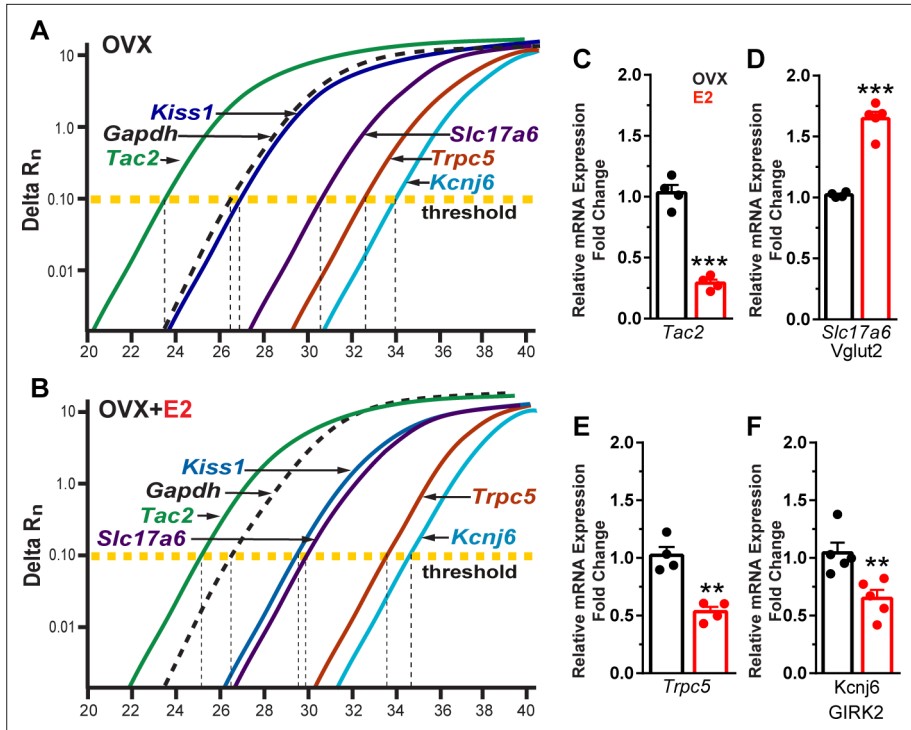

**Figure 10.** Estradiol decreases *Tac 2, Trpc5, and Kcnj6* but increases *Slc17a6* mRNA expression in Kiss1^ARH neurons. (**A**) qPCR amplification curves illustrating the cycle threshold (CT) for *Tac2, Gapdh, Kiss1, Trpc5, Slc17a6* (*Vglut2*), *and Kcnj6* (*Girk2*) in Kiss1^ARH ten cell neuronal pools (3–6 pools from each animal) in OVX Oil-treated and (**B**) OVX E2-treated females. (**C**) Quantitative real-time PCR analysis of *Tac2* mRNA, (**D**) *Slc17a6*, (**E**) *Trpc5*, and (**F**) *Kcnj6*. Comparisons were made between oil-treated and E2-treated, OVX females using the comparative $2^{-\Delta\Delta CT}$ method. Bar graphs represent the mean ± SEM, with data points representing individual animals (unpaired *t*-test for *Tac2*, $t_{(6)}$ = 10.670, ***p<0.0001; for *Slc17a6*, $t_{(7)}$ = 9.678, ***p<0.0001; for *Trpc5*, $t_{(6)}$ = 5.774, **p=0.0012; unpaired *t*-test *for Kcnj6, $t_{(8)}$ = 3.457, **p=0.0086).

The online version of this article includes the following source data for figure 10:

**Source data 1.** Data presented in *Figure 10*.

## Discussion

E2 appears to play a critical role in transitioning the glutamatergic/peptidergic Kiss1^ARH neurons from a high-frequency firing mode during synchronization, which is dependent on NKB-driven activation of TRPC5 channels, to a short bursting mode that would facilitate glutamate release. E2 decreased the expression of the peptide neurotransmitters NKB (kisspeptin and dynorphin) and TRPC5 channels but increased the mRNA expression of *Slc17a6,* voltage-activated calcium channels and *Hcn* channels that contribute to burst firing and glutamate release from the Kiss1^ARH neurons. We determined that the increase in mRNA expression of the HVA calcium channels translated into a significant increase in whole-cell current with all of the calcium channels contributing proportionally. Most importantly the kinetics of activation and inactivation were unaltered with E2 treatment, which indicates that other post-translation modifications were not affecting channel activity. Surprisingly and somewhat coun-terintuitive, the *BK α1* subunit was also upregulated, but based on our modeling the rapid repolariza-tion of the Kiss1^ARH neurons (i.e., the fast AHP) would facilitate higher frequency of action potential firing. Moreover, our modeling confirmed that TRPC5 channels, which generate the slow EPSP (a.k.a., plateau potential in other CNS neurons), are vital for initiating and sustaining synchronous firing of Kiss1^ARH neurons, while subsequent activation of GIRK channels repolarizes Kiss1^ARH neurons. E2 treat-ment of ovariectomized females decreased both *Trpc5* and *Kcnj6* channel mRNA expression, which in our model correlated with the reduction in sustained high-frequency firing of Kiss1^ARH neurons. Therefore, the synchronous high-frequency firing of Kiss1^ARH neurons in a low E2 milieu correlates with the pulsatile release of GnRH (LH from the pituitary gland), whereas the transition to burst firing in the

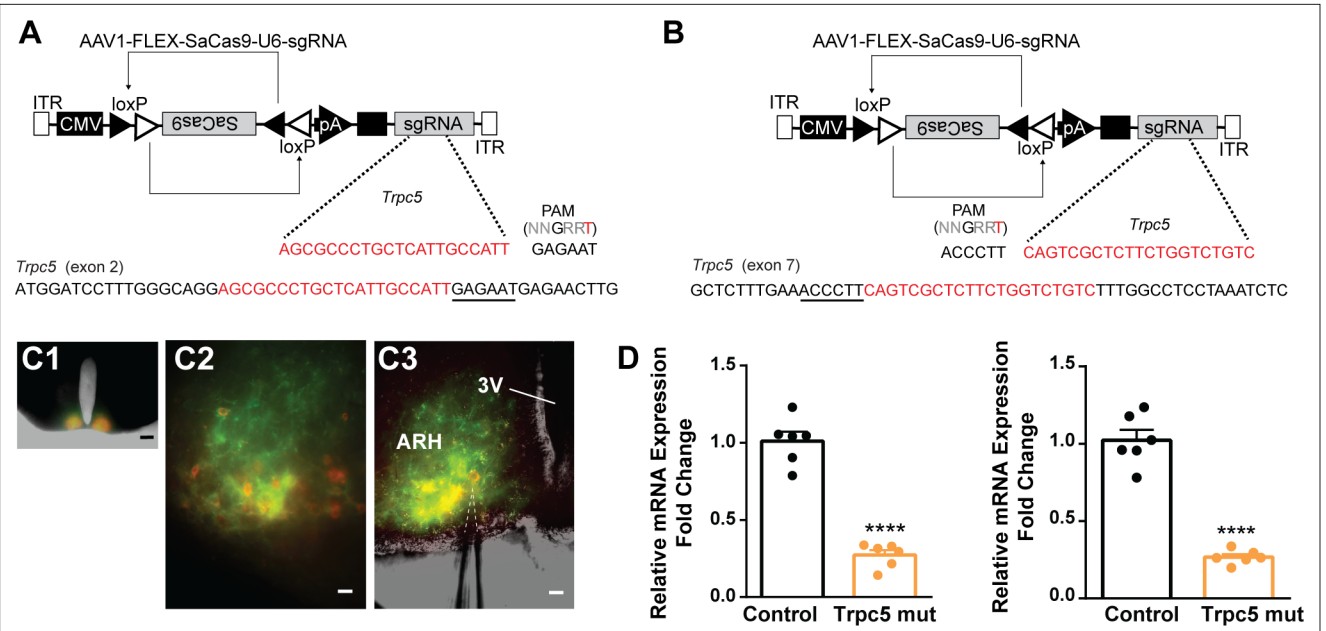

**Figure 11.** CRISPR mutagenesis of *Trpc5* channels in Kiss1[ARH] neurons. (**A**) Structure of AAV1-FLEX-SaCas9-U6sg*Trpc5-exon2*. Exon 2 of *Trpc5* is denoted with guide sequence highlighted in red, the PAM is underlined. (**B**) Structure of AAV1-FLEX-SaCas9-U6sg*Trpc5-exon7*. Exon 7 of *Trpc5* is denoted with guide sequence highlighted in red, the PAM is underlined. (**C1**) Image of coronal section through the ARH from Kiss1-Cre::Ai32 mouse with dual co-injections of AAV-DIO-mCherry and AAV1-FLEX-SaCas9-U6-sg*Trpc5*. Scale = 200 μm. (**C2, C3**) Higher-power overlays of epifluorescence (EYFP and mCherry) images with recording pipette patched onto a Kiss1[ARH]-Cre:mCherry cell (**C3**). Scale = 40 μm. (**D**) Quantitative PCR measurements of *Trpc5* transcripts from 10-cell neuronal pools (three pools from each animal) in double sgRNA mutagenesis of *Trpc5* (second sgRNA against pore-forming region) in Kiss1[ARH] neurons. Primers were targeted to first (left panel) or second (right panel) guide, respectively (unpaired *t*-test for first, $t_{(10)}$ = 10.67, ****p<0.0001; for second, $t_{(10)}$ = 10.79, ****p<0.0001). Data are expressed as mean ± SEM, with data points representing individual animals.

The online version of this article includes the following source data for figure 11:

**Source data 1.** Data presented in *Figure 11*.

## Core calcium conductances underlying synchronous and burst firing of Kiss1[ARH] neurons

TRPC5 channels are highly expressed in Kiss1[ARH] neurons (*Figure 11*), and TRPC5 channels are essentially ligand-activated calcium channels with a high permeability to calcium ($P_{Ca}/P_{Na}$ = 9:1) (*Venkatachalam and Montell, 2007*). In general, mammalian TRPC channels are activated by both G protein-coupled receptors and receptor tyrosine kinases (*Clapham, 2003*; *Ambudkar and Ong, 2007*), and are one of the major downstream effectors activated by glutamate binding to group I metabotropic glutamate receptors (mGluR1 and mGluR5) in CNS neurons (*Tozzi et al., 2003*; *Bengtson et al., 2004*; *Faber et al., 2006*; *Berg et al., 2007*). In substantia nigra dopamine neurons, mGluR1 agonists induce a current that exhibits the tell-tale double-rectifying current–voltage plot of TRPC channel activation (*Tozzi et al., 2003*), similar to what we see with the effects of the NKB agonist senktide in Kiss1[ARH] neurons (*Qiu et al., 2021*). Both mGluR1 and TacR3 are Gq-coupled to phospholipase C (PLC) activation, which leads to hydrolysis of phosphatidylinositol 4,5-bisphosphate ($PIP_2$) to inositol 1,4,5 triphosphate ($IP_3$) and diacylglycerol (DAG), which is involved in channel activation (*Birnbaumer, 2009*). TacR3 (NKB) signaling has additional consequences since many $K^+$ (e.g., GIRK, KCNQ) channels are dependent on PIP2 for channel opening, and depletion of PIP2 by PLC leads to channel closure (*Brown and Passmore, 2009*; *Zhang et al., 2013a*; *Whorton and MacKinnon, 2011*; *Zheng et al., 2022*). Therefore, depletion of PIP2 by NKB signaling would further facilitate the sustained firing of Kiss1[ARH] neurons during synchronization.

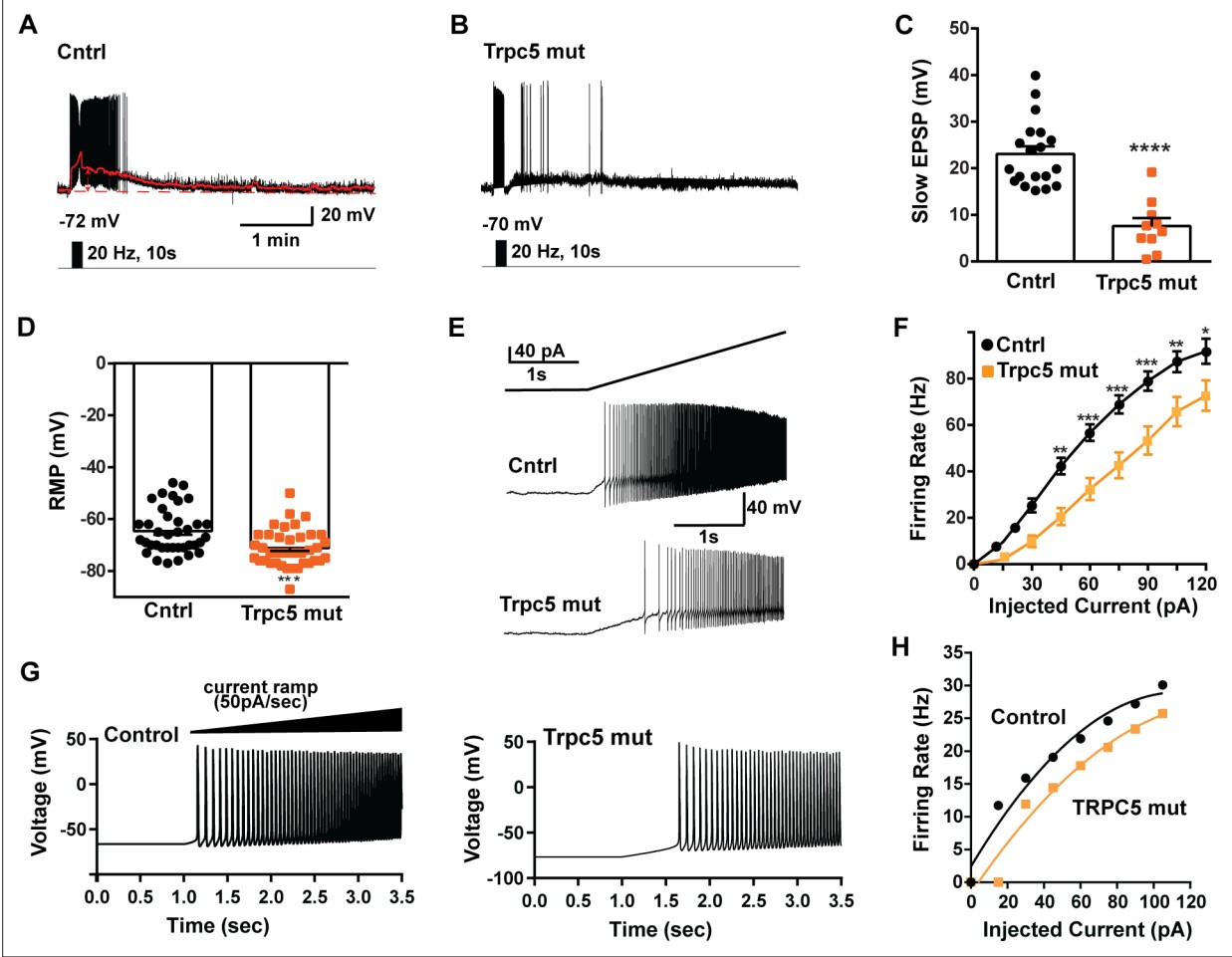

**Figure 12.** Double CRISPR mutagenesis of *Trpc5* attenuates slow excitatory postsynaptic potential (EPSP), increases rheobase, and shifts the F-I curve. (**A**) High-frequency photostimulation (20 Hz) generated slow EPSP in Kiss1ARH neuron from ovariectomized, control mouse. Red trace is slow EPSP after low-pass filtering. (**B**) Slow EPSP in Kiss1ARH neuron from OVX, double sg*Trpc5*-targeted (*Trpc5* mut) mouse. (**C**) Summary of the effects of *Trpc5* mutagenesis on slow EPSP amplitude in female mice. Unpaired *t*-test, $t_{(27)}$ = 5.916, ****p<0.0001. Data are expressed as mean ± SEM, with data points representing individual cells. (**D**) Double sgRNA mutagenesis of *Trpc5* channels in Kiss1ARH neurons significantly increased the RMP (control: –64.7 ± 1.4 mV versus *Trpc5* mut: –71.1 ± 1.2 mV, unpaired *t*-test, $t_{(73)}$ = 3.524, ***p=0.0007). (**E**) Current ramp showing the increased rheobase in a Kiss1ARH neuron from *Trpc5* mut mice (control: 31.1 ± 1.2 pA, n = 31, versus *Trpc5* mut, 35.3 ± 1.0 pA, n = 33, unpaired *t*-test, $t_{(62)}$ = 2.777, **p=0.0073). (**F**) Firing frequency vs. current (F-I) curves for control versus *Trpc5* mut (two-way ANOVA: main effect of treatment [$F_{(1,52)}$ = 13.04, p=0.0007], main effect of injected current [$F_{(8,416)}$ = 291.3, p<0.0001] and interaction [$F_{(8,416)}$ = 6.254, p<0.0001]; control, n = 26, *Trpc5* mut, n = 28; post hoc Bonferroni test, *p<0.05; **p<0.01, and ***p<0.005, respectively). (**G**) Model simulations of the effects of a current ramp (50 pA/s) for OVX (left panel) and OVX female with reduced (muted) TRPC5 conductance (right panel), and (**H**) the associated firing frequency versus current curves. In the latter case, the TRCP5 conductance was halved, which is a conservative estimation of the CRISPR state in which the *Trpc5* is much more mutated in Kiss1ARH neurons (*Figure 11D*).

The online version of this article includes the following source data for figure 12:

**Source data 1.** Data presented in *Figure 12*.

---

A plateau potential has been characterized in hippocampal and cortical neurons (*Zhang et al., 2011*; *Arboit et al., 2020*) as such neurons express biophysical properties that allow them to continue to persistently fire even after a triggering synaptic event has subsided (*Zylberberg and Strowbridge, 2017*). The persistent firing activity of these neurons is linked to $I_{CAN}$ (*Zylberberg and Strowbridge, 2017*), and TRPC5 channels appear to be responsible for the $I_{CAN}$ (*Zhang et al., 2011*). With TacR3 activation in Kiss1ARH neurons, there is an influx of $Ca^{2+}$ through TRPC5 channels, leading to greater build-up of $[Ca^{2+}]_i$ that facilitates the opening of more TRPC5 channels in a self-sustaining (autocatalytic) manner (*Qiu et al., 2016*). Using the fast intracellular calcium chelator BAPTA, which has been shown to robustly inhibit TRPC5 channel activation in heterologous cells (*Blair et al., 2009*), we abolished

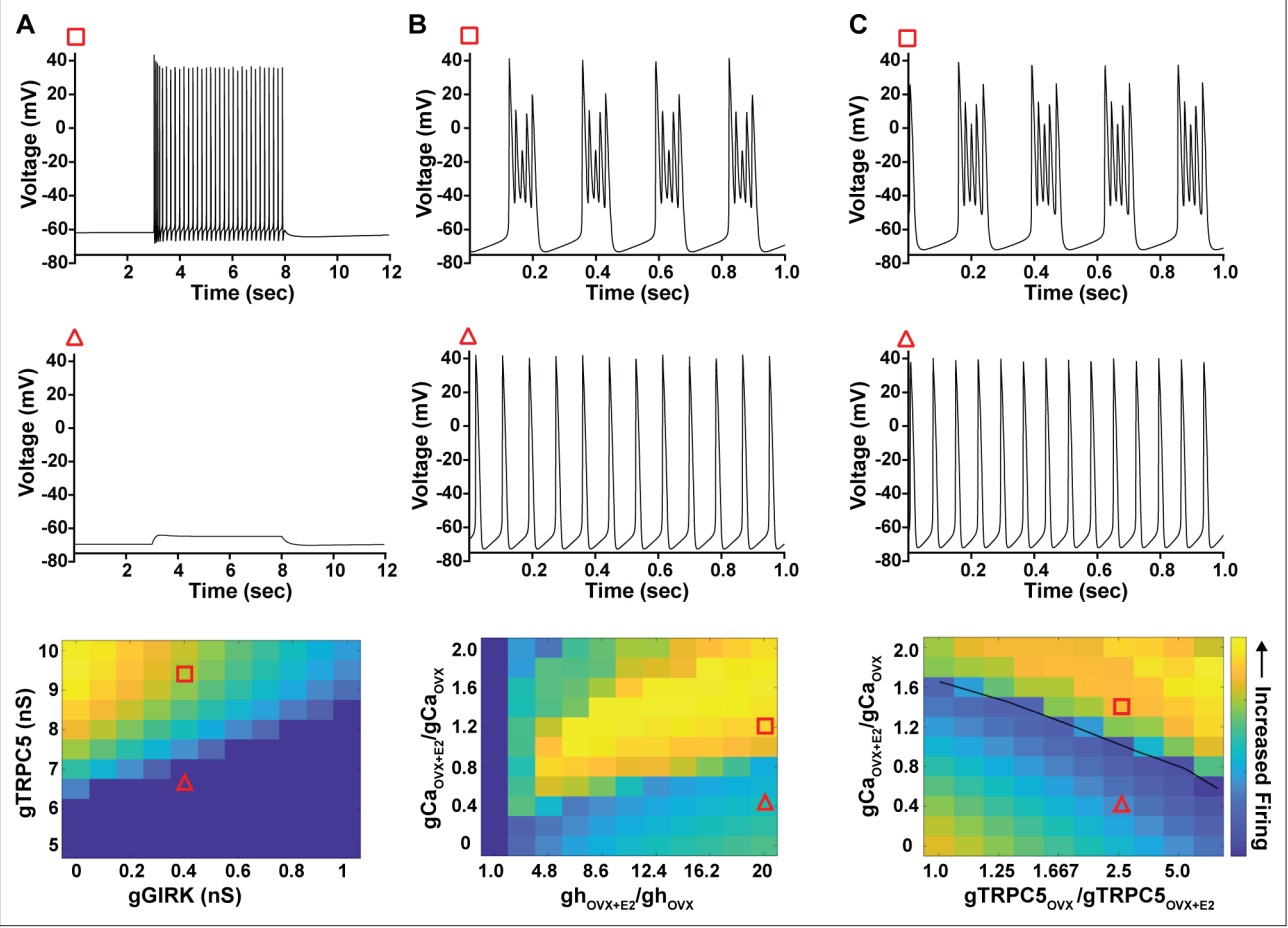

**Figure 13.** Computational modeling of a Kiss1[ARH] neuron in the OVX and OVX + E2 state demonstrates its distinct dynamic responses. A model of the Kiss1[ARH] neuron was developed and calibrated using molecular data and electrophysiological recordings of Kiss1[ARH] neurons from OVX and OVX + E2 mice. (**A**) Simulations of the OVX-parameterized model demonstrating high-frequency activity in response to saturating levels of NKB stimulation. The balance between GIRK and TRCP5 conductance controls the response of the neuron to NKB stimulation, with neuronal response eliminated when TRPC5 conductance is low (red triangle) relative to the GIRK conductance. (**B**) The OVX + E2 parameterized models demonstrate sustained burst firing activity. The bursting activity that is supported by elevated h- and Ca²⁺ currents (red square) as observed in OVX + E2 mice. (**C**) In the OVX + E2 state, burst firing activity is also supported by high conductance of HVA Ca²⁺ channels relative to the conductance of TRPC5 channels. Representative points in the parameter space giving rise to burst firing activity are marked with red squares, whereas red triangles are used for points resulting in regular spiking. The black line separates these two regions of activity.

The online version of this article includes the following source data for figure 13:

**Source data 1.** Data presented in *Figure 13*.

the slow EPSP and persistent firing of Kiss1[ARH] neurons following optogenetic stimulation (*Qiu et al., 2021*). Moreover, HVA calcium channel blockers attenuated the generation of the slow EPSP (*Figure 3*) so it appears that they also contribute to the $I_{CAN}$ since calcium influx via both LVA and HVA calcium channels can also facilitate TRPC5 channel opening in Kiss1[ARH] neurons. In the ovariectomized female, treatment with E2 upregulated *Cacna1c, Cacna1a, Cacna1b,* and *Cacna1e* mRNA by 1.5- to 2-fold and *Cacna1g* mRNA expression by ~3-fold. Hence, E2 significantly increased whole-cell calcium currents in Kiss1[ARH] neurons, which greatly enhanced the excitability and contributed to the burst firing of Kiss1[ARH] neurons (*Figure 1*). However, the amplitude of the slow EPSP with E2 treatment is only ~25% of the amplitude in the ovariectomized state (*Qiu et al., 2016*). Therefore, there appears to be a physiological transition of Kiss1[ARH] neurons from the slow EPSP firing mode in the OVX state to the burst firing mode in the presence of E2, which has important physiological ramifications as discussed below.

## TRPC5 and GIRK channels are vital for synchronization of Kiss1[ARH] neurons

Recently, Tian et al. demonstrated that TRPC4, a close homolog of TRPC5 sharing ~64% homology, is a 'coincidence detector' of neurotransmission by both Gq/11 and Gi/o-coupled receptors in lateral septal (LS) neurons (*Tian et al., 2022*). In whole-cell recordings of LS neurons, TRPC 4 channels mediate a strong depolarizing plateau potential that, in contrast to TRPC5 channel activation in Kiss1[ARH] neurons, abrogates action potential firing as a result of a depolarization block. In many instances, the plateau potential in LS neurons is followed by an AHP, which is dependent on the activation of Gi/o-coupled receptors. In contrast, we have not observed an AHP in Kiss1[ARH] neurons following the slow EPSP (*Qiu et al., 2016*). Tian and colleagues showed that the depolarizing plateau in LS neurons is codependent on the activation of both Gq/11-coupled mGluR1 glutamate receptors and Gi/o-coupled γ-aminobutyric acid type B (GABA_B) receptors, the latter activating GIRK channels. Moreover, the firing patterns in LS neurons encode information about the relative strengths of these contrasting inputs (i.e., Gq/11 versus Gi/o) such that only mGluR1 produces weak depolarization accompanied by increased firing of LS neurons, whereas pure GABA_B receptor activation hyperpolarizes the cells and abrogates firing activity. Coincident input of both mGluR1 and GABA_B receptors results in a brief burst of action potentials followed by a pause in firing, and both the pause duration and firing recovery patterns reflect the relative strengths of Gq/11 versus Gi/o inputs. Importantly, Tian and colleagues simulated these various scenarios with computational modeling, and similar to our modeling, utilized only TRPC4 and GIRK channels. A notable difference between the Kiss1[ARH] neurons and the LS neurons is that dynorphin binds to kappa-opioid receptors to open GIRK channels in the nerve terminal of Kiss1[ARH] neurons (presynaptic effect) (*Qiu et al., 2016*), whereas GABA binds to soma GABA_B receptors to hyperpolarize LS neurons (*Tian et al., 2022*). Therefore, although the high-frequency firing activity of both LS and Kiss1[ARH] neurons can be modeled around TRPC and GIRK channels, the generated firing patterns are dramatically different based on the timing (coincident activation of TRPC4 and GIRK channels in LS) and localization of the kappa opioid-coupled GIRK channels in the axon terminal of Kiss1[ARH] neurons. Moreover, E2-treated, ovariectomized females show a significant downregulation of both *Trpc5* and *Kcnj6* mRNA expression in Kiss1[ARH] neurons (*Figure 10*), which is important for the physiological transitioning as described below.

Interestingly, the hypothalamic A12 (ARH) dopamine neurons show a rhythmic 'oscillatory' firing behavior that transitions to a tonic firing mode with synaptic input from thyrotropin-releasing hormone (TRH) neurons (*Lyons et al., 2010*) or feedback by circulating prolactin, released by pituitary lactotrophs (*Lyons et al., 2012*). A more recent paper has revealed that TRPC5 channels mediate the plateau potential and tonic firing in A12 dopamine neurons in response to prolactin (*Blum et al., 2019*). Similar to our findings with CRISPR deletion of *Trpc5* in Kiss1[ARH] neurons (*Figures 11 and 12*), conditional knockout of *Trpc5* in dopamine neurons abrogated the prolactin-induced plateau potential and tonic firing. Although Blum and colleagues did not model the oscillatory firing or tonic firing of the A12 dopamine neurons, their findings are consistent with our results showing that the activation of TRPC5 channels underlies the slow EPSP (plateau potential) and sustained firing.

## Contribution of endogenous K⁺ channels to synchronized and burst firing

Beyond the ligand-gated (e.g., baclofen) GIRK channels, there are endogenous K⁺ channels that help sculpt the firing activity of kisspeptin neurons. We focused on the calcium-activated K⁺ channel family: the large-conductance, calcium-activated potassium (BK, also called BK_Ca, K_Ca1.1, MaxiK, *Slo*), small conductance, calcium-activated K⁺ (SK1, SK2, SK3) (*Bond et al., 2005*), and the K⁺ channels underlying the M current (KCNQ, Kv7.1–7.5) (*Brown and Passmore, 2009*), which mediate the fAHP, the intermediate AHP/slow AHP, respectively (*Andrade et al., 2012*).

BK channels are gated by both voltage and cytoplasmic calcium and sculpt action potential firing in CNS neurons (*Blatz and Magleby, 1987*; *Marty, 1989*; *Storm, 1990*; *Sah, 1996*). Indeed, BK channels have been shown to mediate rapid spike repolarization—that is, the fast AHP in hippocampal CA1 pyramidal neurons (*Lancaster and Nicoll, 1987*; *Storm, 1987*). Blockade of BK channels in CA1 neurons attenuates the initial discharge frequency in response to current injection, which is attributable to suppression of the BK channel-dependent rapid spike repolarization (*Lancaster and Nicoll, 1987*; *Storm, 1987*). Blockade of BK channels is thought to increase inactivation of the spike-generating

transient Na$^+$ current and activate more of the slower K$^+$ currents, thereby enhancing refractoriness and reducing excitability during the immediate aftermath of the first action potential (*Shao et al., 1999*). Thus, BK channels facilitate high-frequency burst firing of CA1 neurons. Furthermore, extracellular field recordings confirmed that BK channels contribute to high-frequency burst firing in response to excitatory synaptic input to distal dendrites in CA1 neurons (*Gu et al., 2007*). Therefore, BK channels appear to play an important role for early high-frequency, rapidly adapting firing in hippocampal CA1 pyramidal neurons, thus promoting the type of bursting that is characteristic of these cells in vivo during behavior. Based on our in vitro electrophysiological recordings and computational modeling, we see a similar physiological phenomenon in Kiss1$^{ARH}$ neurons (*Figure 8*). Not only does E2 increase the mRNA expression of *Kcnma1* (*Figure 8G*), but also the maximum BK (IbTx-sensitive) current by 3-fold, which would facilitate rapid repolarization during burst firing and promote glutamate release similar to hippocampal CA1 neurons.

In contrast to BK channel expression, E2 did not affect the mRNA expression of *Kcnn3* channel mRNA. SK channels underlie the apamin-sensitive component of the medium duration AHP and are responsible for repolarization following a burst of action potentials (*Bond et al., 2005*; *Andrade et al., 2012*). The activation of SK channels is voltage-independent, but SK channels have a higher affinity for Ca$^{2+}$ than BK channels (*Andrade et al., 2012*). SK channels are tightly coupled (within 100 nm) to L-type Ca$^{2+}$ channels, and BK channels are tightly coupled (within 30 nm) to N-type Ca$^{2+}$ channels in hippocampal CA1 pyramidal neurons (*Marrion and Tavalin, 1998*). The determination of the proximity of SK and BK channels to HVA calcium channels in kisspeptin neurons will require cell-attached patch recordings. However, in Kiss1$^{ARH}$ neurons, the SK channels may come into play during a short burst of action potentials but would become overwhelmed with the higher frequency synchronized, sustained firing as a result of NKB stimulation (*Qiu et al., 2016*). As discussed above, what limits the synchronized firing of Kiss1$^{ARH}$ neurons is the activation of GIRK channels.

Finally, the calcium-activated slow AHP probably plays a critical role in the repolarization of Kiss1$^{ARH}$ neurons after burst firing. The molecular identification of the channels mediating the slow AHP has long been an area of intense investigation (*Vogalis et al., 2003*; *Andrade et al., 2012*). A critical feature of the slow AHP is that it activates very slowly (hundreds of milliseconds) long after the rise in cytoplasmic Ca$^{2+}$ (*Sah and Clements, 1999*) so an intermediate Ca$^{2+}$ signaling molecule has long been thought to be involved. Indeed, *Tzingounis et al., 2007* have provided compelling evidence that the diffusible calcium sensor hippocalcin is the critical intermediate molecule involved in Ca$^{2+}$ sensing. The slow AHP is abrogated in hippocalcin KO mice, and transfection of hippocalcin into cultured hippocampal neurons generates a pronounced slow AHP in response to a depolarizing stimulus (*Tzingounis et al., 2007*). Importantly, the slow AHP is activated by Ca$^{2+}$ with an EC$_{50}$ $\approx$ 300 nM, which is well within the operational range of hippocalcin but well below that of calmodulin (*Andrade et al., 2012*). Finally, two seminal papers from Tzingounis and colleagues demonstrate that KCNQ 2, 3 channels are responsible for the slow AHP in hippocampal dentate neurons (*Tzingounis and Nicoll, 2008*), and KCNQ 5 channels are responsible for the slow AHP in CA3 neurons (*Tzingounis et al., 2010*). Moreover, the KCNQ channel blocker XE991 attenuates the slow AHP in CA3 neurons (*Tzingounis and Nicoll, 2008*). Based on these seminal findings, we investigated the role of the KCNQ channels, which 'classically' underlie the M current in Kiss1$^{ARH}$. The M current was first identified in Kiss1$^{ARH}$ neurons by *Conde and Roepke, 2020*, and presently we found that *Kcnq2* mRNA is expressed in Kiss1$^{ARH}$ neurons and upregulated by E2, which translated into a greater M current in Kiss1$^{ARH}$ neurons (*Figure 9*). Incorporating the M current into our computational model indeed supports our hypothesis that M current, along with I$_{SK}$ and I$_{BK}$, contributes to membrane repolarization after burst firing (*Figure 9G*). Importantly the sAHP, as opposed to the fAHP (BK) and mAHP (SK), is highly regulated by multiple neurotransmitters (*Andrade et al., 2012*), which sets the stage for further modulation of the slow EPSP in Kiss1$^{ARH}$ neurons.

## Importance of E2-driven physiological transitioning

Since the expression of the peptide neurotransmitters in Kiss1$^{ARH}$ neurons is downregulated by E2, the Kiss1$^{ARH}$ neurons are believed to be under 'inhibitory' control by E2 and are important for 'negative-feedback' regulation of GnRH release (*Smith et al., 2005*; *Navarro et al., 2009*; *Lehman et al., 2013*; *Rance, 1991*; *Rance, 2009*). However, our past (*Gottsch et al., 2011*) and current findings document that these Kiss1$^{ARH}$ neurons express HVA and LVA calcium and HCN (pacemaker) channels

and are excited by co-released glutamate from neighboring Kiss1[ARH] neurons, which indicates that these neurons have pacemaker electrophysiological properties similar to other CNS neurons (**Bal and McCormick, 1993**; **Lüthi and McCormick, 1998**). Additionally, in contrast to the neuropeptides, E2 increases *Slc17a6* (*Vglut2*) mRNA expression, in addition to mRNA for the voltage-activated calcium and HCN channels, and increases glutamate release onto Kiss1[AVPV/PeN] neurons (**Qiu et al., 2018**). Interestingly, *Slc17a6* mRNA expression in Kiss1[ARH] neurons and the probability of glutamate release are decreased along with the neuropeptides in intact versus castrated males (**Nestor et al., 2016**), which indicates a profound sex difference in the glutamate signaling by Kiss1[ARH] neurons (**Nestor et al., 2016**; **Qiu et al., 2018**). Obviously, in the male there is no preovulatory LH surge so there is no need for excitatory glutamatergic input to the few Kiss1[AVPV/PeN] neurons in the male.

In females, conditional knockout of *Slc17a6* in Kiss1 neurons abrogates glutamate release from Kiss1[ARH] neurons (**Qiu et al., 2018**). Kiss1[AVPV/PeN] neurons do not express *Slc17a6* and do not release glutamate. Within the Kiss1[ARH] neurocircuitry, the lack of glutamate transmission does not diminish the slow EPSP in ovariectomized females (**Qiu et al., 2018**). Indeed, a recent publication from the Herbison lab demonstrates that glutamate generates small 'synchronizing' events that are dependent on the ionotropic receptors (**Han et al., 2023**), but the fast neurotransmitter is unable to support the sustained firing (i.e., slow EPSP) that is necessary for peptide release and synchronization of the KNDy network. Rather, we postulate that glutamate neurotransmission is more important for excitation of Kiss1[AVPV/PeN] neurons and facilitating the GnRH (LH) surge with high circulating levels of E2 when peptide neurotransmitters are at a nadir and glutamate levels are high in female Kiss1[ARH] neurons. Indeed, low-frequency (5 Hz) optogenetic stimulation of Kiss1[ARH] neurons, which only releases glutamate in E2-treated, ovariectomized females (**Qiu et al., 2016**), generates a surge-like increase in LH release during periods of optical stimulation (**Lin et al., 2021**; **Voliotis et al., 2021**). In a subsequent study, optical stimulation of Kiss1[ARH] neuron terminals in the AVPV at 20 Hz, a frequency commonly used for terminal stimulation in vivo, generated a similar surge of LH (**Shen et al., 2022**). Additionally, intra-AVPV infusion of glutamate antagonists, AP5 + CNQX, completely blocked the LH surge induced by Kiss1[ARH] terminal photostimulation in the AVPV (**Shen et al., 2022**). Therefore, there appears to be a clear role for glutamatergic transmission from the Kiss1[ARH] to Kiss1[AVPV/PeN] neurons in amplifying the LH surge in the female mouse. Finally, it is important to keep in mind that even in the presence of high physiological levels of E2, the mRNA expression of *Tac2* is many fold higher than *Kiss1* (**Figure 10**), which is essential for NKB maintaining synchronous firing of Kiss1[ARH] neurons, albeit at a lower frequency, across all physiological states (**Qiu et al., 2016**). Indeed, there is a progressive change from a strictly pulsatile pattern of GnRH in the hypophyseal portal circulation to one containing both pulsatile and non-pulsatile components during the development of the GnRH surge in the ewe (**Evans et al., 1995a**; **Evans et al., 1995b**), and a pulsatile mode of LH secretion during the preovulatory LH surge is also evident in other species including humans (**Rossmanith et al., 1990**). Therefore, we believe that our cellular molecular and electrophysiological findings in combination with our computational modeling provide a foundation for understanding the complex role of Kiss1[ARH] neurons in controlling fertility in the mammal. Finally, our model provides the first comprehensive biophysical description of the conductances underlying the neuronal activity of Kiss1[ARH] neurons, which can serve as a basis for future computational modeling of the Kiss1[ARH] neuronal network and its interactions with other brain regions involved in the complex regulation of mammalian female reproduction.

# Materials and methods

## Key resources table

| Reagent type (species) or resource | Designation | Source or reference | Identifiers | Additional information |
|---|---|---|---|---|
| Strain, strain background (*Mus musculus*) | C57BL/6J | The Jackson Laboratory | RRID:IMSR_JAX:000664 | |
| Genetic reagent (*M. musculus*) | Kiss1[Cre:GFP] version 2 (V2) | Dr. Richard D. Palmiter; University of Washington; PMID:29336844 | RRID:IMSR_JAX:033169 | |
| Genetic reagent (*M. musculus*) | Ai32 | The Jackson Laboratory | RRID:IMSR_JAX:024109 | |

*Continued on next page*

*Continued*

| Reagent type (species) or resource | Designation | Source or reference | Identifiers | Additional information |
|---|---|---|---|---|
| Genetic reagent (*Adeno-associated virus*) | AAV1-FLEX-SaCas9-U6-sgTrpc5-exon2 | Dr. Larry S. Zweifel; University of Washington | | |
| Genetic reagent (*Adeno-associated virus*) | AAV1-FLEX-SaCas9-U6-sgTrpc5-exon7 | Dr. Larry S. Zweifel; University of Washington | | |
| Genetic reagent (*Adeno-associated virus*) | AAV1-FLEX-SaCas9-U6-sgRosa26 | Dr. Larry S. Zweifel; University of Washington | | |
| Genetic reagent (*Adeno-associated virus*) | AAV1-Ef1α-DIO-ChR2:YFP | Dr. Stephanie L. Padilla; University of Washington; PMID:25429312 | | |
| Genetic reagent (*Adeno-associated virus*) | AAV1-Ef1α-DIO-ChR2:mCherry | Dr. Stephanie L. Padilla; University of Washington; PMID:25429312 | | |

## Animals

All the animal procedures described in this study were performed in accordance with institutional guidelines based on National Institutes of Health standards and approved by the Institutional Animal Care and Use Committee at Oregon Health and Science University (OHSU).

## Mice

*Kiss1^Cre* transgenic female mice version 2 (*Padilla et al., 2018*) were selectively bred at OHSU. They also were crossed with heterozygous Ai32 mice (RRID:IMSR_JAX:024109, C57BL/6 background), which carry ChR2 (H134R)–EYFP gene in their Gt(ROSA)26Sor locus (*Madisen et al., 2012*). All animals were maintained under controlled temperature and photoperiod (lights on at 0600 hr and off at 1800 hr) and given free access to food (Lab Diets 5L0D) and water. Where specified, Kiss1^Cre mice received viral injections to express channelrhodopsin 2 (ChR2) in Kiss1^ARH neurons, 14–21 days prior to each experiment as described (*Qiu et al., 2016*). Some of the females were ovariectomized 7 days prior to an experiment. Each animal was injected on day 5 following OVX with 0.25 µg E2 or vehicle, followed on day 6 with 1.50 µg E2 or vehicle and used for experiments on day 7 (*Bosch et al., 2013*).

## AAV delivery to Kiss1^Cre mice

Fourteen to twenty-one days prior to each experiment, the *Kiss1^Cre* mice (>60 days old) received bilateral ARH injections of a Cre-dependent adeno-associated viral (AAV; serotype 1) vector encoding mCherry (AAV1-Ef1α-DIO-mCherry) or AAV1 vectors designed to encode SaCas9 and single-guide RNAs (sgRNAs) (see the SaCas9 section for specifics on the sgRNA design). Using aseptic techniques, anesthetized female mice (1.5% isoflurane/O₂) received a medial skin incision to expose the surface of the skull. The glass pipette with a beveled tip (diameter = 45 µm) was filled with mineral oil, loaded with an aliquot of AAV using a Nanoject II (Drummond Scientific). ARH injection coordinates were anteroposterior (AP): −1.20 mm, mediolateral (ML): ±0.30 mm, dorsoventral (DV): −5.80 mm (surface of brain z = 0.0 mm); 500 nl of the AAV ($2.0 \times 10^{12}$ particles/ml) was injected (100 nl/min) into each position, and the pipette left in place for 10 min post-injection, then slowly retracted from the brain. The skin incision was closed using Vetbond (3 M) and each mouse received analgesia (Rimadyl, 4–5 mg/kg, s.c.).

## Generation of AAV1-FLEX-SaCas9-U6-sg*Trpc5*

The generation of AAV1-FLEX-SaCas9-U6-sg*Trpc5* viruses was done at the University of Washington using published methods (*Gore et al., 2013*; *Hunker et al., 2020*). The constructs of sgRNAs for *Trpc5* (AAV1-FLEX-SaCas9-U6-sg*Trpc5*) were designed to target exon 2 and exon 7, respectively (*Figure 11A and B*). To achieve *Trpc5* mutagenesis in Kiss1^ARH neurons, Kiss1^Cre mice were co-injected with AAV1-DIO-mCherry, AAV1-FLEX-SaCas9-U6-sg*Trpc5*-exon2, and AAV1-FLEX-SaCas9-U6-sg*Trpc5*-exon7 at a ratio of 10, 45, and 45%, respectively. Control animals were co-injected with AAV1-FLEX-SaCas9-U6-sg*Rosa26* and AAV1-DIO-mCherry at a ratio of 90 and 10%, respectively. AAV1-DIO-mCherry was co-injected with Cas9 vectors to confirm the targeting of injections and visualize the infected Kiss1^ARH

neurons. Each animal was used for experiments 3 weeks after the viral injection, with OVX performed 1 week prior to the experiments.

## Visualized whole-cell patch recording

Electrophysiological and optogenetic studies were made in coronal brain slices (250 μm) containing the ARH from AAV1-EF1α-DIO-mCherry injected Kiss1Cre:GFP or Kiss1-Cre:EYFP::AI32 mice, which were vehicle-treated OVX, and E2-treated OVX females 10 weeks and older as previously described (*Qiu et al., 2016*; *Qiu et al., 2018*). Whole-cell patch recordings were performed in voltage-clamp and current-clamp as previously described (*Qiu et al., 2018*) using an Olympus BX51 W1 fixed-stage scope outfitted with epifluorescence and IR-DIC video microscopy. Patch pipettes (A-M Systems; 1.5 μm outer diameter borosilicate glass) were pulled on a Brown/Flaming puller (Sutter Instrument, model P-97) and filled with the following solution (unless otherwise specified): 128 mM potassium gluconate, 10 mM NaCl, 1 mM $MgCl_2$, 11 mM EGTA, 10 mM HEPES, 2 mM ATP, and 0.25 mM GTP adjusted to pH 7.3 with KOH; 295 mOsm. Pipette resistances ranged from 3.5 to 4 MΩ. In whole-cell configuration, access resistance was less than 30 MΩ; the access resistance was 80% compensated. The input resistance was calculated by measuring the slope of the I-V relationship curve between −70 and −50 mV. Standard whole-cell patch recording procedures and pharmacological testing were performed as previously described (*Qiu et al., 2003*; *Qiu et al., 2014*). Electrophysiological signals were digitized with a Digidata 1322A (Axon Instruments), and the data were analyzed using p-Clamp software (Molecular Devices, Foster City, CA). The liquid junction potential was corrected for all data analysis.

For optogenetic stimulation, a light-induced response was evoked using a light-emitting diode (LED) 470 nm blue light source controlled by a variable 2A driver (ThorLabs, Newton, NJ) with the light path directly delivered through an Olympus ×40 water-immersion lens. For high-frequency (20 Hz) stimulation, the length of stimulation was 10 s (*Qiu et al., 2016*).

For studying the firing pattern of Kiss1[ARH] neurons from OVX and E2-treated OVX mice, the electrodes were filled with an internal solution consisting of the following (in mM): 120 $KMeSO_4$, 5 $Na_2$ phosphocreatine, 10 KCl, 2 $MgSO_4$, 10 HEPES, 4 $K_2ATP$, and 0.25 GTP adjusted to pH 7.3 with KOH; 290 mOsm. The bath solution as described (*Ashhad and Feldman, 2020*) consisted of (in mM) 124 NaCl, 3.5 KCl, 25 $NaHCO_3$, 0.5 $NaH_2PO_4$ and 30 D-glucose,1 $MgSO_4$, 1.5 $CaCl_2$, gassed with 95% $O_2$/5% $CO_2$ (pH 7.4, 310 mOsm).

For studying the activation/inactivation characteristics of the $Ca^{2+}$ current, the electrodes were filled with an internal solution as described (*Qiu et al., 2003*) consisting of the following (in mM): 100 $Cs^+$ gluconate, 20 TEA-Cl, 10 NaCl, 1 $MgCl_2$, 10 HEPES, 11 EGTA, 1 ATP, 0.25 GTP, the pH was adjusted to 7.3 with CsOH at 300 mOsm. The bath solution as described (*Zhang and Spergel, 2012*) consisted of (in mM) 117.5 NaCl, 25 $NaHCO_3$, 1.25 $NaH_2PO_4$, 10 TEA-Cl, 2 $CaCl_2$, 1 $MgCl_2$, 20 sucrose and 5 glucose, gassed with 95% $O_2$/5% $CO_2$ (pH 7.4, 309 mOsm) and supplemented with 1 μM TTX, 10 μM CNQX, 50 μM AP5, and 100 μM picrotoxin. The effects of estrogen treatment on the peak current, peak current density, and activation/inactivation characteristics of calcium current were measured. Activation curves were fitted by the Boltzmann equation: $I/I_{max} = 1/\{1 + \exp[V_{1/2} − Vs/k]\}$, where I is the peak current at the step potential Vs, $I_{max}$ is the peak current amplitude, $V_{1/2}$ is the step potential yielding half-maximum current, and k is the slope factor. Inactivation curves were fit with the Boltzmann equation: $I/I_{max} = 1 − 1/\{1 + \exp[(V_H − V_{1/2})/k]\}$, where I is the peak current at the step potential $V_H$, $I_{max}$ is the peak current amplitude, $V_{1/2}$ is the step potential at which half the current is inactivated, and k is the slope factor.

To record M currents, pipettes were filled with an internal solution consisting of 10 mM NaCl, 128 mM K-gluconate, 1 mM MgCl, 10 mM HEPES, 1 mM ATP, 1.1 mM EGTA, and 0.25 mM GTP (pH 7.3; 290 mOsm). During voltage-clamp, we employed a standard deactivation protocol (*Roepke et al., 2011*; *Conde and Roepke, 2020*) to measure potassium currents. This involved 500 ms voltage steps ranging from –30 to –75 mV in 5 mV increments, following a 300 ms prepulse to –20 mV. The amplitude of the M current relaxation or deactivation was quantified as the difference between the initial (<10 ms) and sustained current (>475 ms) of the current trace.

The bath solution for whole-cell recording of BK, SK, and M currents was artificial cerebrospinal fluid (aCSF) supplemented with 1 μM TTX, 10 μM CNQX, 50 μM AP5, and 100 μM picrotoxin.

## Electrophysiological solutions/drugs

A standard aCSF was used (*Qiu et al., 2003*; *Qiu et al., 2010*). All drugs were purchased from Tocris Bioscience unless otherwise specified. 1 mM TTX (Alomone Labs), 50 mM DL-2-amino-5-phosphonopentanoic acid sodium salt (AP5), 10 mM 6-cyano-7-nitroquinoxaline-2,3-dione disodium (CNQX), 100 mM picrotoxin, 10 mM nifedipine (Sigma), 2 mM $\omega$-conotoxin GVIA (ConoGVIA; Alomone Labs), 0.5 mM $\omega$-conotoxin MVIIC (ConoMVIIC; Alomone Labs), 50 µM $\omega$-agatoxin IVA (AgaIVA; Alomone Labs), 50 µM SNX-482 (Alomone Labs), 10 mM TTA-P2 (TTAP2; Alomone Labs), 200 mM $CdCl_2$ ($Cd^{2+}$; Sigma), 40 mM XE991 (Alomone Labs), 100 µM Iberiotoxin (Alomone Labs), 500 µM Apamin (Alomone Labs), 100 mM glutamic acid, and 1 mM senktide. Stocks (1000×) were prepared in dimethylsulfoxide (DMSO) (picrotoxin, TTAP2, and senktide) or water (TTX, AP5, CNQX, ConoGVIA, ConoMVIIC, AgaIVA, SNX-482, $CdCl_2$, and glutamic acid) and stored at −20°C. Aliquots of the stock solutions were stored as appropriate until needed.

## Cell harvesting of dispersed Kiss1[Cre] neurons and real-time quantitative PCR (qPCR)

Cell harvesting and qPCR was conducted as previously described (*Bosch et al., 2013*). The ARH was microdissected from basal hypothalamic coronal slices obtained from female Kiss1[Cre] version 2 mice (*Padilla et al., 2018*) (n = 4–6 animals/group). The dispersed cells were visualized, patched, and then harvested (10 cells/tube) as described previously (*Bosch et al., 2013*). Briefly, ARH tissue was incubated in papain (7 mg/ml in oxygenated aCSF) for 50 min at 37°C then washed four times in low $Ca^{2+}$ aCSF and two times in aCSF. For cell dispersion, Pasteur pipettes were flame polished to decreasing tip sizes and gentle trituration used to disperse the neurons onto a glass-bottom dish. The plated cells were bathed in oxygenated aCSF using a peristaltic pump to keep the cells viable and clear of debris. Healthy cells with processes and a smooth cell membrane were harvested. Pipettes (World Precision Instruments; 1.5 µm outer diameter borosilicate glass) were pulled on a Brown/Flaming puller (Sutter Instrument, model P-87) to a 10 µm diameter tip. The cells were harvested using the XenoWorks Microinjector System (Sutter Instruments, Navato, CA) which provides negative pressure in the pipette and fine control to draw the cell up into the pipette. Cell pools were harvested and stored at –80°C. All cell pools were DNAse-treated using DNase1. cDNA synthesis was performed as previously described (*Bosch et al., 2013*).

Primers for the genes that encode for low and high voltage-activated $Ca^{2+}$ channels, *Trpc5*, *Slc17a6*, *Kcnma1* (BKα1, large conductance calcium-activated $K^+$ channel), *Kcnn3* (SK3, small conductance calcium-activated $K^+$ channel), and *Gapdh* were designed using Clone Manager software (Sci Ed Software) to cross at least one intron-exon boundary and optimized as previously described using Power SYBR Green method (*Bosch et al., 2013*). Real-time qPCR controls included neuronal pools without reverse transcriptase (-RT), hypothalamic RNA with RT (+) and without RT (-), as well as water blanks. Standard curves using ARH cDNA were utilized to determine the real-time PCR efficiency ($E = 10^{(−1/m)}$ – 1) (*Pfaffl, 2001*; *Biosystems, 2006*). Only primers resulting in efficiencies of 90–100% were used for analysis. Primer sequences, qPCR parameters, and efficiency calculations are provided in *Table 1*.

## mRNA expression analysis

qPCR was performed on a Quantstudio 7 Flex Real-Time PCR System (Applied Biosystems) using Power SYBR Green PCR Master Mix (Applied Biosystems) according to established protocols (*Bosch et al., 2013*). The comparative $\Delta\Delta C_T$ method (*Livak and Schmittgen, 2001*; *Pfaffl, 2001*; *Schmittgen and Livak, 2008*) was used to determine values from duplicate samples of 4 µl for the target genes and 2 µl for the reference gene *Gapdh* in a 20 µl reaction volume containing 1× Power SYBR Green PCR Master Mix and 0.5 µM forward and reverse primers. Three to six 10-cell pools per animal were analyzed from 4 to 6 animals per group. The relative linear quantity was determined using the $2^{-\Delta\Delta CT}$ equation (*Livak and Schmittgen, 2001*; *Pfaffl, 2001*; *Schmittgen and Livak, 2008*). Relative mRNA expression level of target genes in Kiss1[Cre] neurons was obtained by comparing OVX oil-treated controls to OVX E2-treated animals. The mean $\Delta$ CT for the target genes from the OVX oil-treated control samples was used as the calibrator. The data were expressed as *n*-fold change in gene expression normalized to the reference gene *Gapdh* and relative to the calibrator.

## Experimental design and statistical analysis

For the whole-cell patch recording, pharmacological experiments, only one cell was recorded per slice. Two to three slices were analyzed from each Kiss1[Cre] mouse, with at least 3–5 mice contributing to each group. For cell harvesting of dispersed Kiss1[Cre]-YFP neurons and qPCR measurements, 10 cells per pool and 3–6 pools from each animal were used, unless otherwise specified. Statistical comparisons between two groups were performed using an unpaired two-tailed Student's $t$-test. Comparisons between more than two groups were performed using the repeated-measures, multifactorial ANOVA. If a significant interaction was encountered, we then moved to the one-way ANOVA, followed by the multiple range tests as specified in the appropriate figure legends. All data were analyzed using GraphPad Prism version 6. All data are presented as mean ± standard error of the mean (SEM). Differences were considered statistically significant if the probability of error was less than 5%.

## Mathematical model and simulation of neuronal behavior

A mathematical model of the Kiss1[ARH] neuron was developed and calibrated based on our physiological findings. Here, we employed the Hodgkin–Huxley modeling approach (*Hodgkin and Huxley, 1952*). Accordingly, the equation describing the membrane potential $V_m$ of the Kiss1[ARH] neuron is given by

$$C_m \frac{dV_m}{dt} = -I,$$

where $C_m$ is the membrane capacitance and $I$ is the sum of 12 ionic currents:

$$I = I_{NaT} + I_{NaP} + I_A + I_{BK} + I_h + I_{SK} + I_M + I_T + I_{Ca} + I_{TRPC5} + I_{GRIK} + I_{leak}$$

where $I_{NaT}$ and $I_{NaP}$ are the transient and persistent sodium currents, respectively; $I_A$ represents the A current; $I_M$ represents the M current; $I_{SK}$ and $I_{BK}$ are the potassium currents through the SK and BK channels respectively; $I_h$ the HCN current; $I_T$ the T-type calcium current; $I_{Ca}$ represents other calcium currents (L-, N-, P/Q-, and R-type); $I_{TRPC5}$ represents calcium current through the TRPC5 channel; $I_{GIRK}$ potassium current through the GIRK channels. Finally, $I_{leak}$ represents the contribution of leak currents. We use the Hodgkin–Huxley formalism to model current dynamics and their dependence on the membrane potential. A complete specification of all currents along with the complete table of model parameters can be found in the **Appendix 1** material. Model simulations were conducted in MATLAB using the built-in numerical solver (ode45; based on an explicit Runge–Kutta (4, 5) formula). The MATLAB code is available at https://git.exeter.ac.uk/mv286/kiss1-arcuate-neuron-model (copy archived at *Voliotis, 2024*).

## Acknowledgements

All electrophysiology and molecular biological studies were funded by the National Institutes of Health Grant R01-DK68098 (OKR and MJK, multi-PI). The generation of the *sgRNA*s was funded by National Institutes of Health Grants P30-MH048736 and R01-MH104450 (LSZ). The computational modeling was funded by the BBSRC via grant BB/W005883/1 (KTA and MV) and the EPSRC via grant EP/T017856/1 (KTA). We thank Zoe Plain for her contributions to the modeling. The BBSRC also provided an International Partnership Award, BB/3019978/1, to facilitate collaboration between the UK partners (KOB, XFLi, KTA and MV) and the US partners (MJK, OKR, JQ, and MAB).

## Additional information

### Funding

| Funder | Grant reference number | Author |
| --- | --- | --- |
| National Institutes of Health | R01-DK68098 | Oline K Rønnekleiv<br>Martin J Kelly |
| National Institutes of Health | P30-MH048736 | Larry S Zweifel |

| Funder | Grant reference number | Author |
|---|---|---|
| Biotechnology and Biological Sciences Research Council | BB/W005883/1 | Margaritis Voliotis Krasimira Tsaneva-Atanasova |
| Engineering and Physical Sciences Research Council | EP/T017856/1 | Krasimira Tsaneva-Atanasova |
| Biotechnology and Biological Sciences Research Council | International Partnership Award BB/3019978/1 | Kevin T O'Byrne |
| National Institutes of Health | R01-MH104450 | Larry S Zweifel |

The funders had no role in study design, data collection and interpretation, or the decision to submit the work for publication.

## Author contributions

Jian Qiu, Conceptualization, Data curation, Formal analysis, Investigation, Methodology, Writing - original draft, Writing – review and editing; Margaritis Voliotis, Conceptualization, Data curation, Formal analysis, Funding acquisition, Investigation, Methodology, Writing - original draft, Writing – review and editing; Martha A Bosch, Data curation, Formal analysis, Investigation, Methodology, Writing – review and editing; Xiao Feng Li, Resources, Funding acquisition, Writing – review and editing; Larry S Zweifel, Resources, Data curation, Funding acquisition, Methodology, Writing – review and editing; Krasimira Tsaneva-Atanasova, Kevin T O'Byrne, Conceptualization, Resources, Funding acquisition, Writing – review and editing; Oline K Rønnekleiv, Conceptualization, Resources, Formal analysis, Supervision, Funding acquisition, Investigation, Methodology, Writing - original draft, Writing – review and editing; Martin J Kelly, Conceptualization, Resources, Formal analysis, Supervision, Funding acquisition, Writing - original draft, Project administration, Writing – review and editing

## Author ORCIDs

Jian Qiu (iD) https://orcid.org/0000-0002-4988-8587
Margaritis Voliotis (iD) https://orcid.org/0000-0001-6488-7198
Larry S Zweifel (iD) https://orcid.org/0000-0003-3465-5331
Krasimira Tsaneva-Atanasova (iD) https://orcid.org/0000-0002-6294-7051
Kevin T O'Byrne (iD) https://orcid.org/0000-0002-2548-4182
Oline K Rønnekleiv (iD) https://orcid.org/0000-0003-1841-4386
Martin J Kelly (iD) https://orcid.org/0000-0002-8633-2510

## Ethics

This study was performed in strict accordance with the recommendations from the National Institutes of Health Guide for the care and use of Laboratory Animals. All animal procedures were conducted according to the approved institutional animal care and use committee (IACUC) protocol (#IP00000585) at Oregon Health and Science University. All surgeries were performed using aseptic techniques under isoflurane anesthesia, and every effort was made to minimize suffering.

Reviewer #1 (Public review): https://doi.org/10.7554/eLife.96691.4.sa1
Reviewer #2 (Public review): https://doi.org/10.7554/eLife.96691.4.sa2
Author response https://doi.org/10.7554/eLife.96691.4.sa3

# Additional files

## Supplementary files
• MDAR checklist

## Data availability
All data generated or analyzed during this study are included in the manuscript and the accompanying source data files.

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

## Appendix 1

## A mathematical model of the arcuate nucleus kisspeptin neuron

A schematic diagram of the arcuate nucleus Kiss1 (Kiss1[ARH]) neuron model is shown in *Appendix 1—figure 1*, and parameter values used in the simulations are given in *Appendix 1—table 1*.

The equation describing the membrane potential, $V_m$, of Kiss1[ARH] neurons is given by

$$C_m \frac{dV_m}{dt} = -I,$$

where $C_m$ is the membrane capacitance and $I$ is the sum of 12 ionic currents:

$$I = I_{NaT} + I_{NaP} + I_A + I_{BK} + I_h + I_{SK} + I_M + I_T + I_{Ca} + I_{TRPC5} + I_{GIRK} + I_{leak}$$

where $I_{NaT}$ and $I_{NaP}$ are the transient and persistent sodium currents, respectively; $I_A$ represents the A current; $I_M$ represents the M current; $I_{SK}$ and $I_{BK}$ are the potassium currents through the SK and BK channels respectively; $I_T$ the T-type calcium current; $I_{Ca}$ represents other calcium currents (L-, N-, P/Q-, and R-type); $I_{TRPC5}$ represents calcium current through the TRPC5 channel; $I_{GIRK}$ potassium current through the GIRK channels. Finally, $I_{leak}$ represents the contribution of leak currents.

We use the Hodgkin–Huxley formalism to model current dynamics and their dependence on the membrane potential. Below we detail the equations governing the currents.

**Transient sodium current:**

$$I_{NaT} = g_{NaT} \cdot m_{NaT,\infty}\left(V_m\right) \cdot h_{NaT} \cdot \left(V_m - E_{Na}\right),$$

where $g_{NaT}$ is the maximum conductance, $E_{Na}$ is the sodium reversal potential, and $h_{NaT}$ is the inactivation gating variable that obeys the following equation:

$$\frac{dh_{NaT}}{dt} = \frac{h_{NaT,\infty}\left(V_m\right) - h_{NaT}}{\tau_{h,NaT}}.$$

Parameter $\tau_{h,NaT}$ dictates the timescale of inactivation and $h_{NaT,\infty}\left(V_m\right)$ is the steady-state inactivation function:

$$h_{NaT,\infty}\left(V_m\right) = \frac{1}{1 + e^{-\left(V_m - V_{h,NaT}\right)/k_{h,NaT}}}.$$

Parameter $V_{h,NaT}$ describes the voltage achieving half-maximal inactivation and parameter $k_{h,NaT}$ is the associated scaling function.

Finally, in the current formulation $m_{NaT,\infty}\left(V_m\right)$ is the steady-state activation function given by

$$m_{NaT,\infty}\left(V_m\right) = \frac{1}{1 + e^{-\left(V_m - V_{m,NaT}\right)/k_{m,NaT}}}.$$

The transient sodium channel is modeled using parameter values from the Purkinje neuron (*Fry et al., 2007*). This neuron was chosen as a baseline as it contains the same subunits, that is, NaV1.1-α, NaV1.2-α, and NaV1.6-α (*Fry et al., 2007*), as the transient sodium channel in Kiss1[ARH] neuron (*Zhang et al., 2015*).

**Persistent sodium current:**

$$I_{NaP} = g_{NaP} \cdot m_{NaP,\infty}\left(V_m\right) \cdot h_{NaP} \cdot \left(V_m - E_{Na}\right),$$

where $g_{NaP}$ is the maximum conductance, and $h_{NaP}$ is the corresponding inactivation gating variable that obeys the following equation:

$$\frac{dh_{NaP}}{dt} = \frac{h_{NaP,\infty}\left(V_m\right) - h_{NaP}}{\tau_{h,NaP}};$$

and the steady-state activation and inactivation functions are given by

$$m_{NaP,\infty}\left(V_m\right) = \frac{1}{1 + e^{-\left(V_m - V_{m,NaP}\right)/k_{m,NaP}}},$$

$$h_{NaT,\infty}\left(V_m\right) = \frac{1}{1 + e^{-\left(V_m - V_{h,NaP}\right)/k_{h,NaP}}}.$$

The above description of the persistent sodium current was taken from a model of the GnRH neuron (*Moran et al., 2016*).

**A current:**

$$I_A = g_A \cdot m_A \cdot h_A \cdot \left(V_m - E_K\right)$$

where $g_A$ denotes the maximum conductance, $E_K$ is the potassium reversal potential, and $m_A$ and $h_A$ are the corresponding activation and inactivation gating variables, which are described by the following equations:

$$\frac{dm_A}{dt} = \frac{m_{A,\infty}\left(V_m\right) - m_A}{\tau_{m,A}};$$

$$\frac{dh_A}{dt} = \frac{h_{A,\infty}\left(V_m\right) - h_A}{\tau_{h,A}}.$$

The steady-state activation and inactivation functions are given by

$$m_{A,\infty}\left(V_m\right) = \frac{1}{1 + e^{-\left(V_m - V_{m,A}\right)/k_{m,A}}},$$

$$h_{A,\infty}\left(V_m\right) = \frac{1}{1 + e^{-\left(V_m - V_{h,A}\right)/k_{h,A}}}.$$

The model of the A current was based on Mendonca's model of Kv4 channels (*Mendonça et al., 2016*) as these channels are also found in Kiss1[ARH] neurons (*Mendonça et al., 2018*).

**BK current:**

$$I_{BK} = g_{BK} \cdot b_{BK,\infty}\left(V_m, c\right) \cdot \left(V_m - E_K\right),$$

Here, $g_{BK}$ is the maximum conductance, and $b_{BK,\infty}\left(V_m, c\right)$ is the steady-state activation function that depends on the membrane potential, $V_m$, as well as on the cytosolic calcium concentration, $c$:

$$b_{BK,\infty}\left(V_m, c\right) = \frac{1}{1 + e^{-\left(V_m - V_{BK}(c)\right)/k_{BK}}},$$

$$V_{BK}\left(c\right) = V_{BK,0} - k_{shift} \log \frac{c}{k_{c,BK}}.$$

The model of the BK current was based on the model presented in *Tsaneva-Atanasova et al., 2007*, with the OVX conductance parameter fitted to the current–voltage relationships recorded from Kiss1[ARH] neurons in the absence and presence of the specific BK blocker, iberiotoxin.

**SK current:**

$$I_{SK} = g_{SK} \cdot b_{SK,\infty}\left(c\right) \cdot \left(V_m - E_K\right)$$

where $g_{SK}$ denotes the maximum conductance, and $b_{SK,\infty}\left(c\right)$ is the steady-state activation function, which depends on the cytosolic calcium concentration, $c$:

$$b_{SK,\infty}\left(c\right) = \frac{c^n}{c^n + K_{SK}^n}.$$

The model of the BK current was based on the model presented in *Booth et al., 2016*, with the OVX conductance fitted to the current–voltage relationships recorded from Kiss1[ARH] neurons in the absence and presence of the specific SK blocker, apamin.

**M current:**

$$I_M = g_M \cdot m_M \cdot (V_m - E_K)$$

where $g_M$ denotes the maximum conductance, and $m_M$ is the corresponding activation gating variable:

$$\frac{dm_M}{dt} = \frac{m_{M,\infty}(V_m) - m_M}{\tau_{m,M}}$$

with the steady-state activation function, $m_{M,\infty}(V_m)$, taking the form

$$m_{A,\infty}(V_m) = \frac{1}{1 + e^{-(V_m - V_{m,A})/k_{m,A}}}.$$

The model of the M current was parameterized using the steady-state voltage-clamp measurements from actuate Kiss1 neurons (**Conde and Roepke, 2020**), while for the activation timescale we used the timescale used in a model of the CA1/3 pyramidal cells (**Nowacki et al., 2011**).

**h current:**

$$I_h = g_h \cdot \left[ p_h \cdot m_{h,1} + (1 - p_h) \cdot m_{h,2} \right] \cdot (V_m - E_h),$$

$g_h$ denotes the maximum conductance, and $m_{h,1}$ and $m_{h,2}$ are separate activation gating variables operating on different timescales ($\tau_{m,h,1}$ and $\tau_{m,h,2}$, respectively):

$$\frac{dm_{h,1}}{dt} = \frac{m_{h,1,\infty}(V_m) - m_{h,1}}{\tau_{m,h,1}}$$

$$\frac{dm_{h,2}}{dt} = \frac{m_{h,2,\infty}(V_m) - m_{h,2}}{\tau_{m,h,2}}$$

The corresponding steady-state activation functions are

$$m_{h,1,\infty}(V_m) = \frac{1}{1 + e^{-(V_m - V_{m,h,1})/k_{m,h,1}}},$$

$$m_{h,2,\infty}(V_m) = \frac{1}{1 + e^{-(V_m - V_{m,h,2})/k_{m,h,2}}}.$$

Finally, parameter $p_h$ dictates the relative contribution of and $m_{h,1}$ and $m_{h,2}$ to the total current. This model of the h current is based on the hippocampal CA1 pyramidal neuron (**Booth et al., 2016**), and the conductance was based on recording from arcuate Kiss1 neurons (**Qiu et al., 2018**).

**T-type calcium current:**

$$I_T = g_T \cdot m_{T,\infty} \cdot \left[ p_T \cdot h_{T,1} + (1 - p_T) \cdot h_{T,2} \right] \cdot (V_m - E_{Ca}),$$

where $g_T$ is the maximum conductance, and $h_{T,1}$ and $h_{T,2}$ are separate inactivation gating variables operating on different timescales ($\tau_{h,T,1}$ and $\tau_{h,T,2}$, respectively):

$$\frac{dh_{h,1}}{dt} = \frac{h_{T,1,\infty}(V_m) - h_{T,1}}{\tau_{h,T,1}}$$

$$\frac{dh_{h,2}}{dt} = \frac{h_{T,2,\infty}(V_m) - h_{T,2}}{\tau_{h,T,2}}$$

The corresponding steady-state inactivation functions are

$$h_{T,1,\infty}(V_m) = \frac{1}{1 + e^{-(V_m - V_{h,T,1})/k_{h,T,1}}},$$

$$h_{T,2,\infty}\left(V_m\right) = \frac{1}{1 + e^{-\left(V_m - V_{h,T,2}\right)/k_{h,T,2}}}.$$

The steady-state activation function is given by

$$m_{T,\infty} = \frac{1}{1 + e^{-\left(V_m - V_{m,T}\right)/k_{m,T}}}.$$

Finally, parameter $p_T$ dictates the relative contribution of and $h_{T,1}$ and $h_{T,2}$ to the total current. The model of the T current was based on Kiss1 neurons data presented in *Zhang et al., 2015*; *Wang et al., 2016*; *Qiu et al., 2018*.

**L-, N-, P/Q-, and R-type calcium currents:**

$$I_{Ca} = g_h \cdot m_{Ca} \cdot h_{Ca} \cdot \left(V_m - E_{Ca}\right)$$

where $g_{Ca}$ denotes the maximum conductance, and $m_{Ca}$ and $h_{Ca}$ are the corresponding activation and inactivation gating variables, which are described by the following equations:

$$\frac{dm_{Ca}}{dt} = \frac{m_{Ca,\infty}\left(V_m\right) - m_{Ca}}{\tau_{m,Ca}};$$

$$\frac{dh_{Ca}}{dt} = \frac{h_{Ca,\infty}\left(V_m\right) - h_{Ca}}{\tau_{h,Ca}}.$$

The steady-state activation and inactivation functions are given by

$$m_{Ca,\infty}\left(V_m\right) = \frac{1}{1 + e^{-\left(V_m - V_{m,Ca}\right)/k_{m,Ca}}},$$

$$h_{Ca,\infty}\left(V_m\right) = \frac{1}{1 + e^{-\left(V_m - V_{h,Ca}\right)/k_{h,Ca}}}.$$

Parameters of the model for the HVA calcium channels were calibrated from the current–voltage relationships obtained from Kiss1[ARH] neurons (see *Figures 2–6*).

**TRPC5 current:**

$$I_{TRPC5} = g_{TRPC5} \cdot b_{TRPC5}\left(c, R_{TRPC5,act}\right) \cdot \left(V_m - E_{TRPC5}\right)$$

where $g_{TRPC5}$ denotes the maximum conductance, and $b_{TRPC5}\left(c, R_{TRPC5,act}\right)$ is the activating gating variable that depends both on cytosolic calcium concentration ($c$ and on NKB-mediated activation of an intermediary effector, :

$$b_{TRPC5}\left(c, R\right) = \frac{R_{TRPC5,act}}{1 + e^{-\left(c - c_{TRPC5}\right)/k_{TRPC5}}}$$

The dynamics of $R_{TRPC5,act}$ (activated form of $R_{TRPC5}$) are described by

$$\frac{dR_{TRPC5,act}}{dt} = \left(k_{R_{TRPC5},0} + k_{R_{TRPC5}} \frac{NKB^{n2}}{NKB^{n2} + K_{NKB}^{n2}}\right) \cdot \left(R_{TRPC5,T} - R_{TRPC5,act}\right) - k_{-R_{TRPC5}} \cdot R_{TRPC5,act};$$

where NKB is the extracellular NKB concentration, $k_{R,0}$ is the basal rate of $R_{TRPC5}$ activation, $k_R$ is the maximal rate of $R_{TRPC5}$ activation in the presence of NKB, $k_{-R}$ is the rate of $R_{TRPC5}$ inactivation, and $R_{TRPC5,T}$ is the total concentration of the effector.

**GIRK current:**

$$I_{GIRK} = g_{GIRK} \cdot b_{GIRK}\left(V_m, R_{GIRK,act}\right) \cdot \left(V_m - E_K\right)$$

where $g_{GIRK}$ denotes the maximum conductance, and $b_{GIRK}\left(V_m, R_{GIRK,act}B\right)$ is the activating gating variable that depends on membrane potential, $V_m$, and on external activation of an intermediary effector, $R_{GIRK,act}$:

$$b_{GIRK}\left(V_m, R_{GIRK,act}\right) = m_{GIRK}\left(V_m\right) R_{GIRK,act}\left(s\right)$$

The dynamics of $m_{GIRK}$ are described by

$$\frac{dm_{GIRK}}{dt} = \frac{m_{GIRK,\infty}\left(V_m\right) - m_{GIRK}}{\tau_{GIRK}\left(V_m\right)}$$

where the steady-state activation function and timescale function are given by

$$m_{GIRK,\infty}\left(V_m\right) = \frac{1}{1 + e^{-\frac{V_m - V_{GIRK}}{k_{GIRK,1}}}} + \frac{\alpha}{1 + e^{-\frac{V_m - V_{GIRK}}{k_{GIRK,2}}}}$$

$$\tau_{GIRK}\left(V_m\right) = \frac{1}{\alpha_\tau e^{-\left(V_m/V_{GIRK}\right)} + \beta_\tau e^{-\left(V_m/V_{GIRK}\right)}}$$

The dynamics of $R_{GIRK,act}$ are described by

$$\frac{dR_{GIRK,act}}{dt} = \left(k_{R_{GIRK},0} + k_{R_{GIRK}}\frac{s^{n3}}{s^{n3} + K_s^{n3}}\right) \cdot \left(R_{GIRK,T} - R_{GIRK,act}\right) - k_{-R_{GIRK}} \cdot R_{GIRK,act};$$

where $s$ is the extracellular concentration of the activation signal.

The model and parameters of the GIRK current were taken from *Tian et al., 2022*.

**Leak currents:**

$$I_{leak} = I_{leak,0} + I_{leak,Ca} = g_{leak,1} \cdot \left(V_m - E_K\right) + g_{leak,2} \cdot \left(V_m - E_{Ca}\right)$$

The leak current parameters were calibrated to current–voltage relationships recorded from Kiss1[ARH] neurons in the absence/presence of iberiotoxin (BK blocker) and apamin (SK blocker).

Intracellular calcium dynamics:

Finally, the intracellular calcium dynamics are described via the following equation:

$$\frac{dc}{dt} = -\left(I_{Ca} + I_T + rI_{TRPC5} + I_{leak,Ca}\right) \cdot \gamma - d_{ca} \cdot c$$

where parameter $\gamma$ converts the currents to molecule fluxes, parameter $r$ is the fraction of ions conducted by TRPC5 that are calcium ions and parameter $d_{ca}$ dictates the linear rate at which calcium is depleted or pumped out of the cell.

**Modeling the effects of E2:**

The effect of E2 on ionic currents is modeled as a change in the maximum conductance parameter. For currents $I_h, I_T$, the change is obtained from electrophysiological recordings from arcuate Kiss1 neurons (*Qiu et al., 2018*); for $I_M$, the change is inferred from qPCR data assuming that the conductance is directly proportional to the mRNA expression as we have previously documented (*Zhang et al., 2009*; *Roepke et al., 2011*; *Zhang et al., 2013b*; *Zhang et al., 2015*; *Qiu et al., 2018*; *Qiu et al., 2021*; *Stincic et al., 2021*); for $I_{Ca}$ and $I_{TRPC5}$, the change is inferred from the qPCR data and validated in electrophysiological recordings; finally for $I_{SK}, I_{BK}, I_{leak}$, OVX + E2 conductances are obtained from current–voltage relationships recorded from Kiss1[ARH] neurons in the absence/ presence of iberiotoxin (BK blocker) and apamin (SK blocker). Parameter $C_m$ is calibrated using direct measurements of the membrane conductance both in the OVX and OVX + E2 state. For the simulation presented in *Figure 13*, conductances were varied within ranges that capture the physiological effect of E2.

## Computer simulations

Integration of the differential equations describing the model was carried out in MATLAB R2023b using a standard fourth-order Runge–Kutta method. Parameter fitting was also conducted in MATLAB R2023b using the least-squares curve-fitting method.

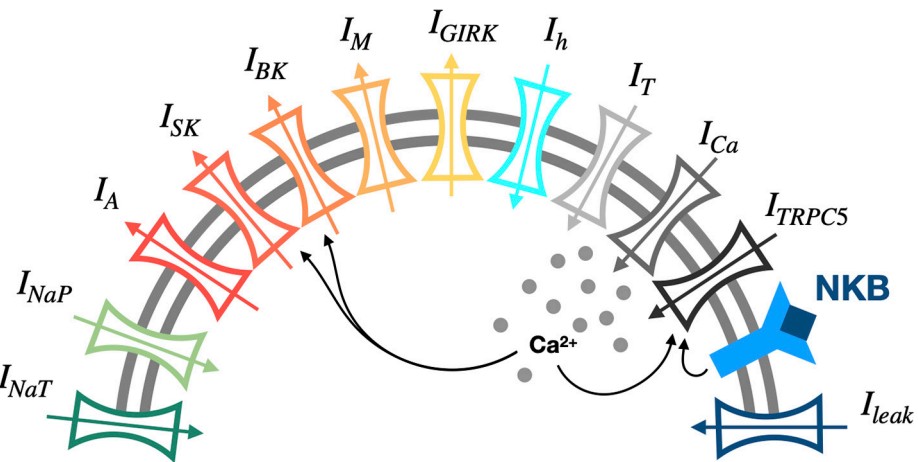

**Appendix 1—figure 1.** Schematic diagram of the conductance-based mathematical model of arcuate nucleus Kiss1 neurons.

**Appendix 1—table 1.** Table of model parameters.

**Model parameters**

| $C_m$ | 23.13 nF (19.5 in OVX + E2) | | | |
|---|---|---|---|---|
| Current parameters* | | | | Reference |
| $I_{NaT}$ | $\tau_{h,NaT}$ | 4.5 ms | $V_{h,NaT}$ | –43.2 mV | *Fry et al., 2007* |
| | $V_{m,NaT}$ | –25 mV | $k_{m,NaT}$ | 3 mV | |
| | $k_{h,NaT}$ | –8.2 mV | $g_{NaT}$ | 90 nS | |
| | $E_{Na}$ | 66.1 mV | | | |
| $I_{NaP}$ | $\tau_{h,NaP}$ | 250 ms | $V_{h,NaP}$ | –47.4 mV | *Zhang et al., 2015*; *Moran et al., 2016* |
| | $V_{m,NaP}$ | –41.5 mV | $k_{m,NaP}$ | 3 mV | |
| | $k_{h,NaP}$ | –8.5 mV | $g_{NaP}$ | 3.37 nS | |
| | $E_{Na}$ | 66.1 mV | | | |
| $I_A$ | $\tau_{h,A}$ | 10 ms | $V_{h,A}$ | –55.1 mV | *Mendonça et al., 2016*; *Mendonça et al., 2018* |
| | $\tau_{m,A}$ | 20 ms | $V_{m,A}$ | –30 mV | |
| | $k_{h,A}$ | –11.4 mV | $k_{m,A}$ | 10 mV | |
| | $g_A$ | 60 nS in OVX 35 in OVX + E2 | $E_K$ | –81 mV | |
| $I_{BK}$ | $k_{BK}$ | 10 mV | $k_{shift}$ | 18 mV | *Tsaneva-Atanasova et al., 2007* |
| | $V_{BK,0}$ | –22.52 mV | $k_{c,BK}$ | 1.5 µM | |
| | $g_{BK}$ | 13.50 nS in OVX; 20 nS in OVX + E2 | $E_K$ | –81 mV | |
| $I_{SK}$ | $K_{SK}$ | 0.45 µM | $n$ | 4 | *Bond et al., 1999* |
| | $g_{SK}$ | 28.13 nS in OVX; 26.05 nS in OVX + E2 | $E_K$ | –81 mV | |

*Appendix 1—table 1 Continued on next page*

*Appendix 1—table 1 Continued*

**Model parameters**

| | | | | | |
|---|---|---|---|---|---|
| $I_M$ | $\tau_{m,M}$ | 10 ms | $V_{m,M}$ | −50 mV | *Nowacki et al., 2011; Conde and Roepke, 2020* |
| | $k_{m,M}$ | 20 mV | $g_M$ | 0.23 nS in OVX 1.23 ns in OVX + E2 | |
| | $E_K$ | −81 mV | | | |
| $I_h$ | $\tau_{m,h,1}$ | 80 ms | $V_{m,h,1}$ | −102 mV | *Gottsch et al., 2011; Booth et al., 2016; Qiu et al., 2018* |
| | $\tau_{m,h,1}$ | 310 ms | $V_{m,h,2}$ | −102 mV | |
| | $p_h$ | 0.85 | $k_{m,h,1}$ | −10 mV | |
| | $k_{m,h,2}$ | −10 mV | $g_h$ | 0.56 nS (11.23 nS in OVX + E2) | |
| | $E_h$ | −27.8 mV | | | |
| $I_T$ | $\tau_{h,T,1}$ | 1.94 ms | $V_{h,T,1}$ | −69.1 mV | *Zhang et al., 2015; Wang et al., 2016* |
| | $\tau_{h,T,2}$ | 86.3 ms | $V_{h,T,2}$ | −69.1 mV | |
| | $p_h$ | 0.5 | $V_{m,T}$ | −54 mV | |
| | $k_{h,T,1}$ | −5.3 mV | $k_{h,T,2}$ | −5.3 mV | |
| | $k_{m,T}$ | 3.3 mV | $g_T$ | 0.66 nS (5 nS in OVX + E2) | |
| $I_{Ca}$ | $\tau_{h,Ca}$ | 10 ms | $V_{h,Ca}$ | −48.9 | fitted |
| | $\tau_{m,Ca}$ | 300 ms | $V_{m,Ca}$ | −27.3 mV | |
| | $k_{h,Ca}$ | −18.2 | $k_{m,Ca}$ | 4.62 mV | |
| | $g_{Ca}$ | 2.1 nS (2.8 nS in OVX + E2) | $E_{Ca}$ | 121.6 mV | |
| $I_{TRPC5}$ | $c_{TRPC5}$ | 0.6 μM | $k_{TRPC5}$ | 0.33 μM | *Qiu et al., 2021* |
| | $k_{R_{TRPC5}}$ | 0.006 ms$^{-1}$ | $k_{-R_{TRPC5}}$ | 0.002 ms$^{-1}$ | |
| | $k_{R_{TRPC5},0}$ | 0.002 ms$^{-1}$ | $g_{TRPC5}$ | 8.4 nS (1.68 nS in OVX + E2) | |
| | $K_{NKB}$ | 32 μM | $n2$ | 2 | |
| | $R_{TRPC5,T}$ | 1 μM | $E_{TRPC5}$ | −15 mV | |
| $I_{GIRK}$ | $V_{GIRK}$ | −70 mV | $k_{GIRK,1}$ | 20 mV | *Tian et al., 2022* |
| | $\alpha$ | 0.8 | $\alpha_\tau$ | 0.0061 | |
| | $k_{GIRK,2}$ | 100 mV | $\beta_\tau$ | 0.0818 | |
| | $g_{GIRK}$ | 0.4 nS | $E_K$ | −81 mV | |
| | $k_{R_{GIRK},0}$ | 0 ms$^{-1}$ | $k_{R_{GIRK}}$ | 10$^{-3}$ ms$^{-1}$ | |
| | $R_{GIRK,T}$ | 1 μM | $K_S$ | 1 μM | |
| | $k_{-R_{GIRK}}$ | 3.3·10$^{-6}$ ms$^{-1}$ | $n3$ | 2 | |
| $I_{leak}$ | $g_{leak,1}$ | 3.5 nS in OVX 3.65 nS in OVX + E2 | $g_{leak,2}$ | 4 nS | Fitted |

Intracellular calcium

*Appendix 1—table 1 Continued on next page*

*Appendix 1—table 1 Continued*

**Model parameters**

| | | | | |
|---|---|---|---|---|
| $\gamma$ | $1.96 \cdot 10^{-5} \mu M\ nA^{-1}\ ms^{-1}$ | $d_{ca}$ | $0.008\ ms^{-1}$ | *Tsaneva-Atanasova et al., 2007* |
| $r$ | 0.9 | | | |

*Parameters $g$. denote the maximum conductance associated with each current.

