## [Editor Report · eLife Assessment]

This **valuable** study combined multiple approaches to gain insight into why rising estradiol levels, by influencing hypothalamic neurons, ultimately lead to ovulation. The experimental data were **solid**, but evidence for the conclusion that the findings explain how estradiol acts in the intact female were incomplete because they lacked experimental conditions that better approximate physiological conditions. Nevertheless, the work will be of interest to reproductive biologists working on ovarian biology and female fertility.

---

## [Referee Report · Reviewer #1 (Public review)]

Summary:

In this work, Qiu and colleagues examined the effects of preovulatory (i.e., proestrous or late follicular phase) levels of circulating estradiol on multiple calcium and potassium channel conductances in arcuate nucleus kisspeptin neurons. Although these cells are strongly linked to a role as the "GnRH pulse generator," the goal here was to examine the physiological properties of these cells in a hormonal milieu mimicking late proestrus, the time of the preovulatory GnRH-LH surge. Computational modeling is used to manipulate multiple conductances simultaneously and support a role for certain calcium channels in facilitating a switch in firing mode from tonic to bursting. CRISPR knockdown of the TRPC5 channel reduced overall excitability, but this was only examined in cells from ovariectomized mice without estradiol treatment.

Comments to address most recent author response:

The concern regarding the CRISPR experiments being confined to OVX mice is that the results can only suggest that CRISPR-mediated knockdown of TRPC5 can, at best, phenocopy the OVX+E condition. A reciprocal experiment in the opposite direction (for example, that returning TRPC5 to OVX levels in OVX+E mice prevents the changes in firing activity and pattern typical of the OVX+E2 condition) would strengthen the indication that E2-sensitive changes in TRPC5 expression and function are critically important to surge function. Acknowledging this as a limitation of the studies would help to better contextualize the value of the CRISPR experiments to an understanding of surge mechanisms when done only in OVX conditions.

The nature of the confusion regarding the consideration of OVX+E2 conditions in the computational model primarily arises from the methods description in the supplemental file: "The effect of E2 on ionic currents is modelled as a change in the maximum conductance parameter. For currents IM,IT, ICa and ITRPC5 this change is inferred from the qPCR data assuming that the conductance is directly proportional to the mRNA expression." If these were instead based on the whole-cell recordings as the authors now indicate in their response, then this description needs to be edited and clarified accordingly. Furthermore, the section states, "For ISK, IBK, Ileak, the OVX and OVX+E2 conductances are obtained from current-voltage relationships recorded from Kiss1ARH neurons in the absence/presence of iberiotoxin (BK blocker) and apamin (SK blocker). All other currents were assumed to be unaffected by E2." This section thus does not directly indicate that the recordings in the stated figures were used in the model, and moreover suggests that currents besides ISK, IBK, and Ileak were not different in OVX+E2 conditions.

The prior evidence stated for correlation of mRNA and channel conductance is not explicitly cited in the manuscript. It is well known that post-translational modifications, physiological modulation of individual channel biophysical properties, and many other factors can influence the end output of a membrane conductance. Therefore, the authors should, at minimum, provide a literature citation supporting the assumption used here.

---

## [Referee Report · Reviewer #2 (Public review)]

Summary:

Kisspeptin neurons of the arcuate nucleus (ARC) are thought to be responsible for the pulsatile GnRH secretory pattern and to mediate feedback regulation of GnRH secretion by estradiol (E2). Evidence in the literature, including the work of the authors, indicates that ARC kisspeptin coordinate their activity through reciprocal synaptic interactions and the release of glutamate and of neuropeptide neurokinin B (NKB), which they co-express. The authors show here that E2 regulates the expression of genes encoding different voltage-dependent calcium channels, calcium-dependent potassium channels and canonical transient receptor potential (TRPC5) channels and of the corresponding ionic currents in ARC kisspeptin neurons. Using computer simulations of the electrical activity of ARC kisspeptin neurons, the authors also provide evidence of what these changes translate into in terms of these cells' firing patterns. The experiments reveal that E2 upregulates various voltage-gated calcium currents as well as 2 subtypes of calcium-dependent potassium currents, while decreasing TRPC5 expression (an ion channel downstream of NKB receptor activation), the slow excitatory synaptic potentials (slow EPSP) elicited in ARC kisspeptin neurons by NKB release and expression of the G protein-associated inward-rectifying potassium channel (GIRK). Based on these results, and on those of computer simulations, the authors propose that E2 promotes a functional transition of ARC kisspeptin neurons from neuropeptide-mediated sustained firing that supports coordinated activity for pulsatile GnRH secretion to a less intense burst-like firing pattern that could favor glutamate release from ARC kisspeptin. The authors suggest that the latter might be important for the generation of the preovulatory surge in females.

Strengths:

The authors combined multiple approaches in vitro and in silico to gain insights into the impact of E2 on the electrical activity of ARC kisspeptin neurons. These include patch-clamp electrophysiology combined with selective optogenetic stimulation of ARC kisspeptin neurons, reverse transcriptase quantitative PCR, pharmacology and CRISPR-Cas9-mediated knockdown of the Trpc5 gene. The addition of computer simulations for understanding the impact of E2 on the electrical activity of ARC kisspeptin cells is also a strength.

The authors add interesting information on the complement of ionic currents in ARC kisspeptin neurons and on their regulation by E2 to what was already known in the literature. Pharmacological and electrophysiological experiments appear of the highest standards and robust statistical analyses are provided throughout. The impact of E2 replacement on calcium and potassium currents is compelling. Likewise, the results of Trpc5 gene knockdown do provide good evidence that the TRPC5 channel plays a key role in mediating the NKB-mediated slow EPSP. Surprisingly, this also revealed an unsuspected role for this channel in regulating the membrane potential and excitability of ARC kisspeptin neurons.

Weaknesses:

The manuscript also has weaknesses that obscure some of the conclusions drawn by the authors.

One is that the authors compare here two conditions, OVX versus OVX replaced with high E2, that may not reflect the physiological conditions under which the proposed transition between neuropeptide-dependent sustained firing and less intense burst firing might take place (i.e. the diestrous [low E2] and proestrous [high E2] stages of the estrous cycle). This is an important caveat to keep in mind when interpreting the authors' findings. Indeed, that E2 alters certain ionic currents when added back to OVX females, does not mean that the magnitude of all of these ionic currents will vary during the estrous cycle.

In addition, although the computational modeling indicates a role of the various E2-modulated conductances in causing a transition in ARC kisspeptin neuron firing pattern, their role is not directly tested in physiological recordings, weakening the link between these changes and the shift in firing patterns.

Overall, the manuscript provides interesting information about the effects of E2 on specific ionic currents in ARC kisspeptin neurons and some insights into the functional impact of these changes. However, some of the conclusions of the work, with regard, in particular, to the role of these changes in ion channels and to their implications for the LH surge, are not fully supported by the findings.

---

## [Author Response]

[The following is the authors’ response to the current reviews.]

**Reviewer #1 (Public review):**
Summary:In this work, Qiu and colleagues examined the effects of preovulatory (i.e., proestrous or late follicular phase) levels of circulating estradiol on multiple calcium and potassium channel conductances in arcuate nucleus kisspeptin neurons. Although these cells are strongly linked to a role as the "GnRH pulse generator," the goal here was to examine the physiological properties of these cells in a hormonal milieu mimicking late proestrus, the time of the preovulatory GnRH-LH surge. Computational modeling is used to manipulate multiple conductances simultaneously and support a role for certain calcium channels in facilitating a switch in firing mode from tonic to bursting. CRISPR knockdown of the TRPC5 channel reduced overall excitability, but this was only examined in cells from ovariectomized mice without estradiol treatment.Comments to address most recent author response:The concern regarding the CRISPR experiments being confined to OVX mice is that the results can only suggest that CRISPR-mediated knockdown of TRPC5 can, at best, phenocopy the OVX+E condition. A reciprocal experiment in the opposite direction (for example, that returning TRPC5 to OVX levels in OVX+E mice prevents the changes in firing activity and pattern typical of the OVX+E2 condition) would strengthen the indication that E2-sensitive changes in TRPC5 expression and function are critically important to surge function. Acknowledging this as a limitation of the studies would help to better contextualize the value of the CRISPR experiments to an understanding of surge mechanisms when done only in OVX conditions.

We have noted in the manuscript that “It would be of interest in future experiments to do the reciprocal experiment to see if overexpressing Trpc5 channels in Kiss1ARH neurons from OVX + E2 females restores the RMP and “rescues” the synchronization phenotype.”

The nature of the confusion regarding the consideration of OVX+E2 conditions in the computational model primarily arises from the methods description in the supplemental file: "The effect of E2 on ionic currents is modelled as a change in the maximum conductance parameter. For currents IM,IT, ICa and ITRPC5 this change is inferred from the qPCR data assuming that the conductance is directly proportional to the mRNA expression." If these were instead based on the whole-cell recordings as the authors now indicate in their response, then this description needs to be edited and clarified accordingly. Furthermore, the section states, "For ISK, IBK, Ileak, the OVX and OVX+E2 conductances are obtained from current-voltage relationships recorded from Kiss1ARH neurons in the absence/presence of iberiotoxin (BK blocker) and apamin (SK blocker). All other currents were assumed to be unaffected by E2." This section thus does not directly indicate that the recordings in the stated figures were used in the model, and moreover suggests that currents besides ISK, IBK, and Ileak were not different in OVX+E2 conditions.The prior evidence stated for correlation of mRNA and channel conductance is not explicitly cited in the manuscript. It is well known that post-translational modifications, physiological modulation of individual channel biophysical properties, and many other factors can influence the end output of a membrane conductance. Therefore, the authors should, at minimum, provide a literature citation supporting the assumption used here.

We have re-written the paragraph on “Modelling the effects of E2” in the Supplemental Information (now Appendix 1) to clarify the that the modeling was based on a combination of electrophysiological recordings and the qPCR data presented in this and previous publications. The statement that “all other currents were assumed to be unaffected by E2” was a misstatement and has been deleted. As per the reviewer’s request, we have listed seven publications that document the correlation between the mRNA expression and channel conductance for the various channels. We thank the reviewer for the suggestion.

**Reviewer #2 (Public review):**
Summary:Kisspeptin neurons of the arcuate nucleus (ARC) are thought to be responsible for the pulsatile GnRH secretory pattern and to mediate feedback regulation of GnRH secretion by estradiol (E2). Evidence in the literature, including the work of the authors, indicates that ARC kisspeptin coordinate their activity through reciprocal synaptic interactions and the release of glutamate and of neuropeptide neurokinin B (NKB), which they co-express. The authors show here that E2 regulates the expression of genes encoding different voltage-dependent calcium channels, calcium-dependent potassium channels and canonical transient receptor potential (TRPC5) channels and of the corresponding ionic currents in ARC kisspeptin neurons. Using computer simulations of the electrical activity of ARC kisspeptin neurons, the authors also provide evidence of what these changes translate into in terms of these cells' firing patterns. The experiments reveal that E2 upregulates various voltage-gated calcium currents as well as 2 subtypes of calcium-dependent potassium currents, while decreasing TRPC5 expression (an ion channel downstream of NKB receptor activation), the slow excitatory synaptic potentials (slow EPSP) elicited in ARC kisspeptin neurons by NKB release and expression of the G protein-associated inward-rectifying potassium channel (GIRK). Based on these results, and on those of computer simulations, the authors propose that E2 promotes a functional transition of ARC kisspeptin neurons from neuropeptide-mediated sustained firing that supports coordinated activity for pulsatile GnRH secretion to a less intense burst-like firing pattern that could favor glutamate release from ARC kisspeptin. The authors suggest that the latter might be important for the generation of the preovulatory surge in females.Strengths:The authors combined multiple approaches in vitro and in silico to gain insights into the impact of E2 on the electrical activity of ARC kisspeptin neurons. These include patch-clamp electrophysiology combined with selective optogenetic stimulation of ARC kisspeptin neurons, reverse transcriptase quantitative PCR, pharmacology and CRISPR-Cas9-mediated knockdown of the Trpc5 gene. The addition of computer simulations for understanding the impact of E2 on the electrical activity of ARC kisspeptin cells is also a strength. The authors add interesting information on the complement of ionic currents in ARC kisspeptin neurons and on their regulation by E2 to what was already known in the literature. Pharmacological and electrophysiological experiments appear of the highest standards and robust statistical analyses are provided throughout. The impact of E2 replacement on calcium and potassium currents is compelling. Likewise, the results of Trpc5 gene knockdown do provide good evidence that the TRPC5 channel plays a key role in mediating the NKB-mediated slow EPSP. Surprisingly, this also revealed an unsuspected role for this channel in regulating the membrane potential and excitability of ARC kisspeptin neurons.Weaknesses:The manuscript also has weaknesses that obscure some of the conclusions drawn by the authors.One is that the authors compare here two conditions, OVX versus OVX replaced with high E2, that may not reflect the physiological conditions under which the proposed transition between neuropeptide-dependent sustained firing and less intense burst firing might take place (i.e. the diestrous [low E2] and proestrous [high E2] stages of the estrous cycle). This is an important caveat to keep in mind when interpreting the authors' findings. Indeed, that E2 alters certain ionic currents when added back to OVX females, does not mean that the magnitude of all of these ionic currents will vary during the estrous cycle.

We do know that the slow EPSP, which is generated by TRPC5 channels, tracks beautifully with the steroid state of female mice. Using our E2 treatment paradigm that generates a LH surge in OVX females (left panel in Author response image 1), there is no difference in the amplitude of the slow EPSP in proestrous versus OVX + E2 females (right panel in Author response image 1).

In addition, although the computational modeling indicates a role of the various E2-modulated conductances in causing a transition in ARC kisspeptin neuron firing pattern, their role is not directly tested in physiological recordings, weakening the link between these changes and the shift in firing patterns.

In future experiments we will test directly the physiological contribution of the other E2-modulated conductances in causing the transition in the firing pattern of arcuate Kiss1 neurons using CRISPR/SaCas9 technology as we have documented for the TRPC5 channel (e.g., Figures 11 and 12).

Overall, the manuscript provides interesting information about the effects of E2 on specific ionic currents in ARC kisspeptin neurons and some insights into the functional impact of these changes. However, some of the conclusions of the work, with regard, in particular, to the role of these changes in ion channels and to their implications for the LH surge, are not fully supported by the findings.

[The following is the authors’ response to the previous reviews.]

**Public Reviews:**

**Reviewer #1 (Public Review):**
Summary:In this work, Qiu and colleagues examined the effects of preovulatory (i.e., proestrous or late follicular phase) levels of circulating estradiol on multiple calcium and potassium channel conductances in arcuate nucleus kisspeptin neurons. Although these cells are strongly linked to a role as the "GnRH pulse generator," the goal here was to examine the physiological properties of these cells in a hormonal milieu mimicking late proestrus, the time of the preovulatory GnRH-LH surge. Computational modeling is used to manipulate multiple conductances simultaneously and support a role for certain calcium channels in facilitating a switch in firing mode from tonic to bursting. CRISPR knockdown of the TRPC5 channel reduced overall excitability, but this was only examined in cells from ovariectomized mice without estradiol treatment. The manuscript has been substantially improved from the initial version by the addition of new experiments and clarification of important figures. Importantly, the overlap of data with previous reports from the same group has been corrected.Strengths:(1) Examination of multiple types of calcium and potassium currents, both through electrophysiology and molecular biology.(2) Focus on arcuate kisspeptin neurons during the surge is relatively conceptually novel as the anteroventral periventricular nucleus (AVPV) kisspeptin neurons have received much more attention as the "surge generator" population.(3) The modeling studies allow for direct examination of manipulation of single and multiple conductances, whereas the electrophysiology studies necessarily require examination of each current in isolation. Construction of an arcuate kisspeptin neuron model promises to be of value to the reproductive neuroendocrinology field.Weaknesses:A remaining weakness in this revised version of the manuscript is that the relevance of the CRISPR experiments is still rather tenuous given that the goal is to understand what happens in the estrogen-treatment condition, and these experiments were performed only in OVX mice. Similar concerns reflect that the computational model examining the effect of E2 infers multiple conductances based on qPCR data and an assumption that the conductances are directionally proportional to the level of gene expression, and then tunes these to the current recordings obtained from OVX mice, without a direct confirmation in OVX+E2 conditions that the model parameters accurately reflect the properties of these currents in the presence of estrogen.

We are still puzzled by Reviewer’s concerns about doing the CRISPRing of Trpc5 in the OVX+E2 females. The Trpc5 channel expression is significantly reduced with the E2 treatment (Figure 10E) which we know translates into a minimal slow EPSP (Figure 2, Qiu eLife 2016) and is essentially equivalent to the slow EPSP amplitude in the Trpc5 mutagenesis in the ovariectomized females (Figure 12). TRPC5 channel conductance is already at “rock bottom.” The modeling informs us that such a low TRPC5 conductance will not support a long lasting slow EPSP and sustained firing (Figure 13A).

Also, we respectively point out that we have published a score of papers over the past 20 years showing that the channel conductance does correlate with the mRNA expression (e.g., Qiu et al., eLife 2018). Secondly, the model does take into consideration the OVX + E2 conditions (Figure 13B,C) which is based on the extensive whole-cell recordings presented in Figures 4,5,6,7,8 and 9.

**Reviewer #2 (Public Review):**
Summary:Kisspeptin neurons of the arcuate nucleus (ARC) are thought to be responsible for the pulsatile GnRH secretory pattern and to mediate feedback regulation of GnRH secretion by estradiol (E2). Evidence in the literature, including the work of the authors, indicates that ARC kisspeptin coordinate their activity through reciprocal synaptic interactions and the release of glutamate and of neuropeptide neurokinin B (NKB), which they co-express. The authors show here that E2 regulates the expression of genes encoding different voltage-dependent calcium channels, calcium-dependent potassium channels and canonical transient receptor potential (TRPC5) channels and of the corresponding ionic currents in ARC kisspeptin neurons. Using computer simulations of the electrical activity of ARC kisspeptin neurons, the authors also provide evidence of what these changes translate into in terms of these cells' firing patterns. The experiments reveal that E2 upregulates various voltage-gated calcium currents as well as 2 subtypes of calcium-dependent potassium currents while decreasing TRPC5 expression (an ion channel downstream of NKB receptor activation), the slow excitatory synaptic potentials (slow EPSP) elicited in ARC kisspeptin neurons by NKB release and expression of the G protein-associated inward-rectifying potassium channel (GIRK). Based on these results, and on those of computer simulations, the authors propose that E2 promotes a functional transition of ARC kisspeptin neurons from neuropeptide-mediated sustained firing that supports coordinated activity for pulsatile GnRH secretion to a less intense burst-like firing pattern that could favor glutamate release from ARC kisspeptin. The authors suggest that the latter might be important for the generation of the preovulatory surge in females.Strengths:The authors combined multiple approaches in vitro and in silico to gain insights into the impact of E2 on the electrical activity of ARC kisspeptin neurons. These include patch-clamp electrophysiology combined with selective optogenetic stimulation of ARC kisspeptin neurons, reverse transcriptase quantitative PCR, pharmacology and CRISPR-Cas9-mediated knockdown of the Trpc5 gene. The addition of computer simulations for understanding the impact of E2 on the electrical activity of ARC kisspeptin cells is also a strength.The authors add interesting information on the complement of ionic currents in ARC kisspeptin neurons and on their regulation by E2 to what was already known in the literature. Pharmacological and electrophysiological experiments appear of the highest standards and robust statistical analyses are provided throughout. The impact of E2 replacement on calcium and potassium currents is compelling. Likewise, the results of Trpc5 gene knockdown do provide good evidence that the TRPC5 channel plays a key role in mediating the NKB-mediated slow EPSP. Surprisingly, this also revealed an unsuspected role for this channel in regulating the membrane potential and excitability of ARC kisspeptin neurons.Weaknesses:The manuscript also has weaknesses that obscure some of the conclusions drawn by the authors.One is that the authors compare here two conditions, OVX versus OVX replaced with high E2, that may not reflect the physiological conditions under which the proposed transition between neuropeptide-dependent sustained firing and less intense burst firing might take place (i.e. the diestrous [low E2] and proestrous [high E2] stages of the estrous cycle). This is an important caveat to keep in mind when interpreting the authors' findings. Indeed, that E2 alters certain ionic currents when added back to OVX females, does not mean that the magnitude of all of these ionic currents will vary during the estrous cycle.

Unfortunately, mice are a poor reproductive model since female mice do not have a clear follicular (estradiol-driven) phase distinctive from the luteal (progesterone-driven) phase. Had we utilized a “proestrous” female, we could not with certainty distinguish between the effects of estradiol versus progesterone on the expression of the calcium and potassium channels that were the focus of this study. Therefore, using our physiological model we can state with confidence that “estradiol elicits distinct firing patterns in arcuate nucleus kisspeptin neurons….”

Overall, the manuscript provides interesting information about the effects of E2 on specific ionic currents in ARC kisspeptin neurons and some insights into the functional impact of these changes. However, some of the conclusions of the work, with regard, in particular, to the role of these changes in ion channels and their implications for the LH surge, are not fully supported by the findings.

As we pointed out in the Discussion, the O’Byrne lab has clearly shown the relevance of Kiss1ARH neuronal burst firing and the release of glutamate to its effects on the LH surge:

“Rather, we postulate that glutamate neurotransmission is more important for excitation of Kiss1AVPV/PeN neurons and facilitating the GnRH (LH) surge with high circulating levels of E2 when peptide neurotransmitters are at a nadir and glutamate levels are high in female Kiss1ARH neurons. Indeed, low frequency (5 Hz) optogenetic stimulation of Kiss1ARH neurons, which only releases glutamate in E2-treated, ovariectomized females (Qiu J. et al., 2016), generates a surge-like increase in LH release during periods of optical stimulation (Lin et al., 2021; Voliotis et al., 2021). In a subsequent study optical stimulation of Kiss1ARH neuron terminals in the AVPV at 20 Hz, a frequency commonly used for terminal stimulation in vivo, generated a similar surge of LH (Shen et al., 2022). Additionally, intra-AVPV infusion of glutamate antagonists, AP5+CNQX, completely blocked the LH surge induced by Kiss1ARH terminal photostimulation in the AVPV (Shen et al., 2022).”

**Recommendations for the authors:**

**Reviewer #2 (Recommendations for The Authors):**
The reviewer noted the following in the revised manuscript:- page 6, the authors may consider adding that presynaptic effects of blocking calcium channels on the slow EPSP cannot be fully ruled out. Indeed, the added experiments do indicate that some of the effects can be explained by impaired regulation of TRPC5 channels by calcium influx through calcium channels; however, the senktide-induced current is not fully blocked by the broad-spectrum calcium channel inhibitor cadmium, suggesting that the effect of blocking these channels on the slow EPSP may involve other mechanisms, such as presynaptic effects.

Optogenetic stimulation of all Kiss1ARH neurons induces the release of NKB at “physiological” concentrations, which in turn generates a slow EPSP in the recorded Kiss1ARH neuron. Blocking voltage-gated calcium channels can inhibit the NKB release from presynaptic Kiss1ARH neurons, thereby reducing the amplitude of the slow EPSP. However, in whole-cell recordings of synaptically isolated Kiss1ARH neurons, senktide directly induces a large inward current (Figure 3F), which is generated by the opening of TRPC5 channels (Qiu et al. J. Neurosci 2021). Voltage-gated calcium channels are coupled to the activation of TRPC5 channels (Blair, Kaczmarek and Clapham, J. Gen Physiol 2009), so by blocking voltage-gated calcium channels, cadmium effectively abrogates the facilitating effects of these channels on TRPC5 channel activation and significantly reduces but does not abolish the inward (excitatory) current (Figures 3F-H). We have clarified in the Results (page 6) that the Kiss1ARH neurons were synaptically isolated as depicted in Figures 3F,G.

- page 8, bottom, the mean value given for the apamin-sensitive current amplitude in E2 treated females does not match that plotted on the I/V graph in Figure 7F.

Thank you for pointing out this typographical error, which we have corrected.